# Ceramic-carbon Janus membrane for robust solar-thermal desalination

Yingchao Dong ®[1] ✉, Camille Violet[2], Chunyi Sun[3], Xianhui Li[4], Yuxuan Sun[1], Qingbin Zheng ®[1] ✉, Chuyang Tang ®[5] ✉ & Menachem Elimelech ®[6,7] ✉

The desalination performance of conventional distillation membranes is limited by insufficient stability and energy efficiency, impeding their application in sustainable water production. Herein, we report a ceramic-carbon Janus membrane with solar-thermal functionality for enhanced desalination performance, energy efficiency, and stability for hypersaline water treatment. The feed and permeate sides of this Janus membrane are designed with different properties such as wettability, conductivity, and solar-thermal conversion to enhance performance. We demonstrate that this membrane exhibits higher solar-thermal efficiency (66.8–68.8%) and water flux (3.3–5.1 L m$^{-2}$ h$^{-1}$) than most existing polymeric solar-thermal distillation membranes. Simulation results ascribe enhanced performance to an increased membrane surface temperature, which mitigates temperature polarization and attenuation, thus enhancing the desalination driving force. The nano-carbon membrane surface accelerates water evaporation by inducing a transition from free water to intermediate water with decreased hydrogen bonding and a lower evaporation energy barrier. Water vapor molecules transport through the membrane pores by a combined mechanism of Knudsen diffusion and viscous flow. Even for seawater and hypersaline water, the membrane exhibits stable water flux and salt rejection due to its scaling-resistant surface and stable interfacial temperature. This work provides a strategy for rationally designing next-generation Janus membranes for sustainable water purification.

Water pollution and scarcity have become urgent global challenges due to population growth and industrial development[1,2]. Membrane-based desalination technologies are increasingly relied on to sustainably produce fresh water from unconventional sources such as seawater, brackish groundwater and wastewater[3]. Membrane distillation (MD) is one such technique which uses a hydrophobic membrane to separate water vapor from saltwater at vapor-liquid equilibrium. While saltwater is unable to permeate the hydrophobic membrane, pure

water vapor molecules can permeate, and migrate across the membrane down their partial pressure gradient. Recently, MD has been gaining attention due to its promising advantages, such as ultrahigh salt rejection, near-atmospheric pressure operation, and potential applications in resource and energy recovery[4–6]. In particular, MD processes are more suitable for treating challenging hypersaline waters than other conventional membrane-based processes such as reverse osmosis[7–9]. However, conventional MD

[1]School of Science and Engineering, The Chinese University of Hong Kong, Shenzhen, Guangdong Province, China. [2]Department of Chemical and Environmental Engineering, Yale University, New Haven, CT, USA. [3]School of Water Conservancy and Environment, Jinan University, Jinan, China. [4]Guangdong Provincial Key Laboratory of Water Quality Improvement and Ecological Restoration for Watersheds, School of Ecology, Environment and Resources, Guangdong University of Technology, Guangzhou, China. [5]Department of Civil Engineering, The University of Hong Kong, Pokfulam, Hong Kong, China. [6]Department of Civil and Environmental Engineering, Rice University, Houston, TX, USA. [7]Department of Chemical and Biomolecular Engineering, Rice University, Houston, TX, USA. ✉e-mail: ycdong@cuhk.edu.cn; zhengqingbin@cuhk.edu.cn; tangc@hku.hk; menachem.elimelech@rice.edu

desalination technologies suffer from high energy consumption, temperature polarization, transverse temperature attenuation, membrane fouling and scaling, and long-term stability[10,11].

To address these issues, solar-thermal distillation membranes were proposed, which enable water supply in off-grid or remote areas where environmental sustainability, affordability of investment, and safety of drinking water are required. Solar-thermal distillation membranes combine conventional hydrophobic distillation membranes with solar-thermal materials that convert renewable solar energy into localized thermal energy on the membrane surface[12–16]. For example, carbonaceous materials such as carbon nanotubes were incorporated into distillation membranes because they possess superior light absorption and rapid solar-thermal conversion due to the presence of conjugated π bonds[9,17–19]. However, incorporating solar-thermal materials in distillation membranes can reduce surface porosity and decrease vapor permeability. Furthermore, most existing solar-thermal distillation membranes are limited by inefficient solar-thermal conversion, water treatment performance, and operational stability, and their interfacial evaporation mechanisms are not well understood. These shortcomings motivate us to design next-generation solar-thermal distillation membranes with energy-efficient separation performance and high water permeability.

Herein, we report a Janus solar-thermal ceramic-carbon membrane with a triple-phase membrane interface for efficiently treating more challenging hypersaline waters and seawater. Unlike conventional distillation membranes, our membrane is rationally designed with special Janus structure, superporous surface structure, and superhydrophobic surface, resulting in enhanced solar-thermal conversion, high flux and permeability, and promising desalination performance and stability. Our design leverages the solar conversion capability of nano-carbon to achieve enhanced interfacial evaporation, desalination performance, and stability (Fig. 1). Different from a previous work focusing on structure optimization and conventional MD performance[19], the key motivation of the current work is to investigate heat and mass transfer and interfacial evaporation mechanisms, as well as the desalination performance and stability of the more challenging real seawater and hypersaline brines via solar-thermal MD process. The membrane was thoroughly characterized, and its performance systematically assessed under a wide range of operation conditions. Both macroscopic and microscopic simulation methods such as computational fluid dynamics and molecular dynamics were employed to elucidate the interfacial evaporation mechanism and the heat and mass transfer properties of the membrane. This membrane shows promising energy efficiency, separation performance, and stability for hypersaline water treatment and outperforms most existing polymeric solar-thermal distillation membranes. Our work exemplifies a rational design strategy of equipping MD membranes with solar-thermal functionality without compromising permeability or operational stability in practical water treatment applications.

## Results

### Solar-thermally enhanced membrane interface
Rational design of solar-thermally enhanced distillation membranes is important for sustainable water desalination. A critical design parameter in MD processes is the membrane surface temperature because it determines the water vapor pressure at the membrane interface and thereby the driving force for desalination. Conventional MD suffers from temperature polarization, where evaporating water molecules lower the temperature at the membrane interface, and tangential temperature attenuation, where heat losses result in decreasing temperature in the direction of feed-flow. Herein, a high-performance MD desalination membrane was rationally designed with a solar-thermal carbon nanotube (CNT) layer that maintains high membrane surface temperature by harnessing sustainable solar energy and reducing conductive heat loss. In addition to high solar-thermal conversion, the CNT layer also provides a superhydrophobic and superporous surface for optimal MD performance (Fig. 2).

The solar-thermally enhanced Janus membranes were fabricated by growing a CNT network layer on $Al_2O_3$ ceramic substrates. Hydrophilic $Al_2O_3$ ceramic membranes (water contact angle ~20°, Fig. 2a inset) with high gas permeance ($1421.1 \pm 62.0$ $m^3$ $m^{-2}$ $h^{-1}$ $bar^{-1}$), high water permeance ($10367.8 \pm 596.7$ L $m^{-2}$ $h^{-1}$ $bar^{-1}$) and sufficient strength ($96.4 \pm 3.6$ MPa) (Supplementary Figs. 3–5) were fabricated (Supplementary Figs. 1, 2 and Supplementary Table 1–2). Inherently, such ceramic membranes are not suitable for MD applications since their water liquid entry pressure (LEP) is below the detection limit (≤0.01 MPa) (Fig. 2f). Therefore, a superhydrophobic CNT network layer (~3.5 μm thickness) was grown in situ on the $Al_2O_3$ membrane surface via chemical vapor deposition (CVD), forming a two-sided Janus membrane structure (Fig. 2b–c, Fig. 2e, Supplementary Figs. 6–9). The Raman intensity ratio of G to D bands ($I_G/I_D$) was ~1.2, suggesting the CNT has a high degree of graphitization (Fig. 2g). Unlike ceramic and conventional MD membranes[20,21], this smooth CNT network layer provides performance-enhancing features such as superhydrophobicity (~168°), higher water liquid entry pressure (1.2 bar), and super-porosity ($80.8 \pm 1.8\%$) (Fig. 2a inset, Fig. 2d, Fig. 2f and Supplementary Figs. 10–13). The CNT layer also exhibits strong solar absorption (~98.1%) for wavelengths ranging from 300 to 2400 nm (Fig. 2h), outperforming most solar-thermal materials such as polydopamine-reduced graphene oxide (PDA-rGO), MXene, ferroferric oxide ($Fe_3O_4$), titanium nitride (TiN), polydopamine-carbon nanotube (PDA-CNT), and carbon black (CB), with the exception of cESM-CNT (Fig. 2i and Supplementary Table 3).

The solar thermal properties of the CNT network layer provide localized surface heating at the solid-liquid-gas interface and effectively maintain a high transmembrane temperature gradient[10]. Compared to the $Al_2O_3$ ceramic membrane with low solar absorption, the CNT-coated membrane exhibits excellent solar absorption and enables efficient solar-thermal conversion (Fig. 3a, b). When CNTs are exposed to solar light energy that matches their electronic transition state, delocalized π orbital electrons in the ground state (the highest occupied molecular orbital, HOMO) are excited to π* orbitals (the lowest unoccupied molecular orbital, LUMO)[11,22–24]. Localized heat energy is produced when the excited electrons transition back to the ground state via a lattice vibration mechanism (Fig. 3c). Due to this solar-thermal effect, the ceramic-carbon Janus membrane shows significantly elevated surface temperature compared to the pristine ceramic membrane (Fig. 3d, e, Supplementary Fig. 14). Surface temperature of the Janus membrane increased rapidly from 23.4 °C to 51.0 °C under exposure to one sun (1 kW $m^{-2}$) irradiation for 400 s in air (Fig. 3d), as verified by infrared thermal images (Fig. 3e). When exposed to 3 sun illumination, the Janus membrane surface temperature increased by ~35.2 °C relative to the $Al_2O_3$ membrane (Fig. 3f), demonstrating excellent surface heating behavior.

The ceramic-carbon Janus membrane also exhibits low thermal conductivity which mitigates temperature polarization and reduces conductive heat loss (Fig. 3g)[25]. Although CNTs are intrinsically thermally conductive[26], the fabricated composite membrane shows a low thermal conductivity of ~0.52 W $m^{-1}$ $K^{-1}$, which is lower than most existing ceramic membranes (e.g., $Al_2O_3$, $ZrO_2$, β-Sialon, $γ-Y_2Si_2O_7$) (Fig. 3h and Supplementary Table 4)[27–29]. This is attributed to the low thermal conductivity of the specially designed $Al_2O_3$ substrate, which comprises multi-level pore structures with long finger-like macrovoids (Supplementary Fig. 5). The Janus membrane exhibits similar thermal conductivity as the substrate because the $Al_2O_3$ to CNT thickness ratio is ~99:1. Due to such a specially designed substrate structure with higher porosity, the Janus membrane shows lower thermal conductivity than a conventional $Al_2O_3$ ceramic membrane with homogeneous particulate-packing pore structure (i.e., lower porosity) (Supplementary Fig. 5)[30]. This fabrication process and

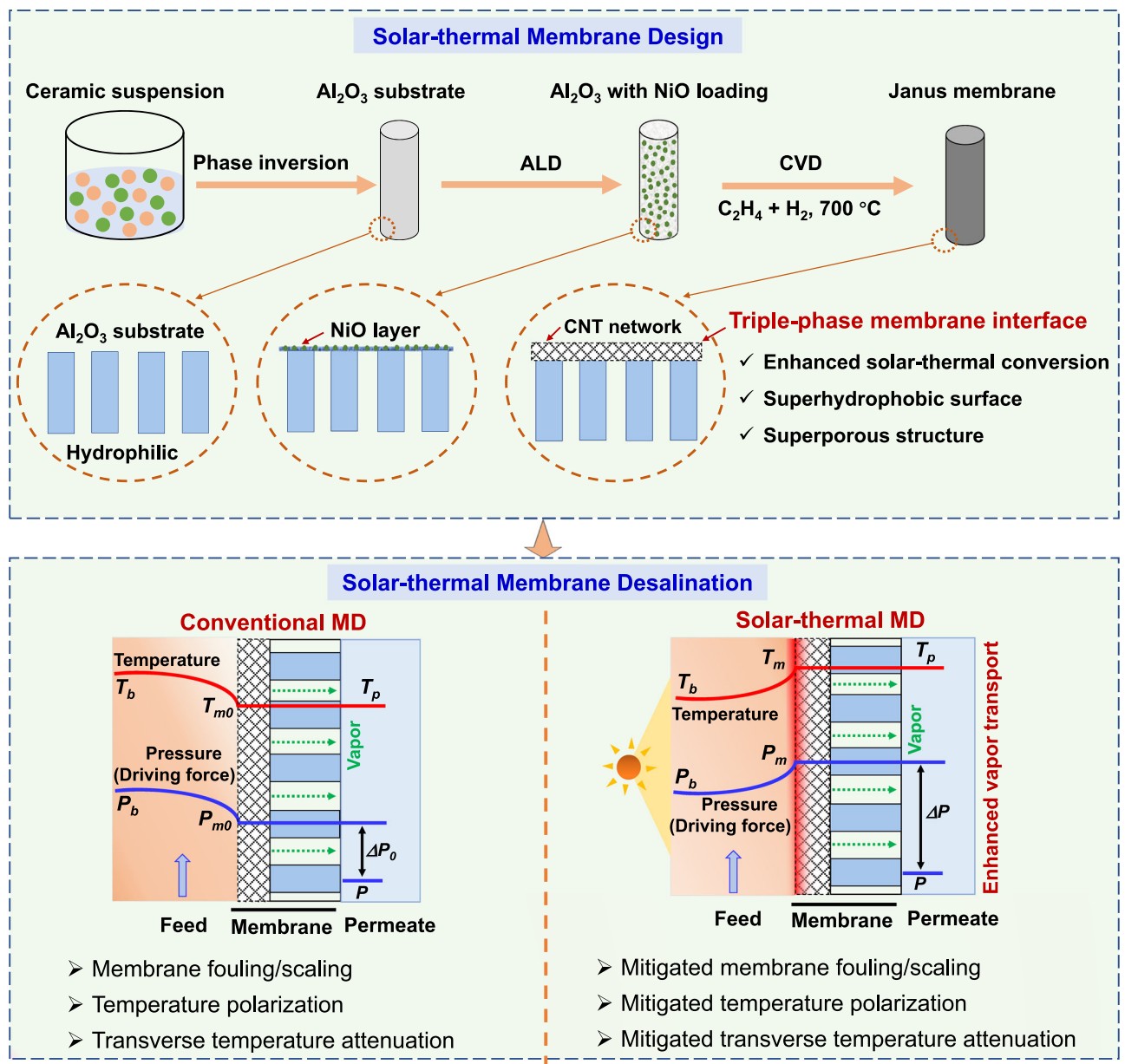

**Fig. 1 | Schematic illustration of membrane design procedures and solar-thermally enhanced desalination process of ceramic-carbon Janus membranes.** The Janus membranes with triple-phase interfaces were fabricated by several procedures including phase inversion, sintering, atomic layer deposition (ALD), and CVD (Al₂O₃: alumina, NiO: nickel oxide, CNT: carbon nanotube). The Janus membranes were then operated in conventional and solar thermal MD processes ($T_f$ and $T_P$ are the bulk temperatures of feed and permeate, respectively. $T_{mO}$ and

$T_m$ represent the feed-membrane interface temperature in conventional MD and solar-thermal MD, respectively. $P_f$ and $P_p$ are the water vapor partial pressures of feed and permeate, respectively. $P_{mO}$ and $P_m$ represent the water vapor partial pressures at the feed-membrane interface in conventional MD and solar-thermal MD, respectively. $\Delta P_O$ and $\Delta P$ represent the water vapor pressure difference between the feed and permeate in conventional MD and solar-thermal MD, respectively).

composite membrane design for low thermal conductivity is expected to be compatible with a variety of substrates (e.g., zirconia, titania, etc). Combined effects of low thermal conductivity and high solar-thermal conversion in the Janus membrane serve to alleviate the temperature polarization, which is an intractable issue in conventional MD.

**Enhanced solar-thermal desalination performance**
Compared with conventional MD processes that require bulk heating to elevate interfacial temperature, emerging solar-thermal MD processes leverage targeted heating at the membrane interface to improve energy utilization, reduce heat loss, and enhance vapor transport[24]. The ceramic-carbon membrane in this work exhibits

promising solar-thermal MD desalination performance as determined by water flux and salt rejection (Fig. 4). Desalination performance was tested and optimized by varying operation parameters such as solar power intensity, feed concentration, feed temperature, and flow rate. Under no solar illumination, the membrane exhibits a low water flux of $7.2 \pm 0.5 \, L \, m^{-2} \, h^{-1}$ but near complete salt rejection (>99.9%) when treating saline water (10 g L⁻¹ NaCl) at room temperature (28 °C) (Fig. 4b). Increasing the solar illumination from 1 to 3 kW m⁻² resulted in increased membrane surface temperature from 32.2 to 37.3 °C (Fig. 4c, d), and thus increased water flux from $8.1 \pm 0.1$ to $9.5 \pm 0.1 \, L \, m^{-2} \, h^{-1}$, while maintaining high salt rejection (>99.9%). Interestingly, even for treatment of highly saline waters (90 g L⁻¹ NaC`l), the solar-thermal membranes showed promising water flux

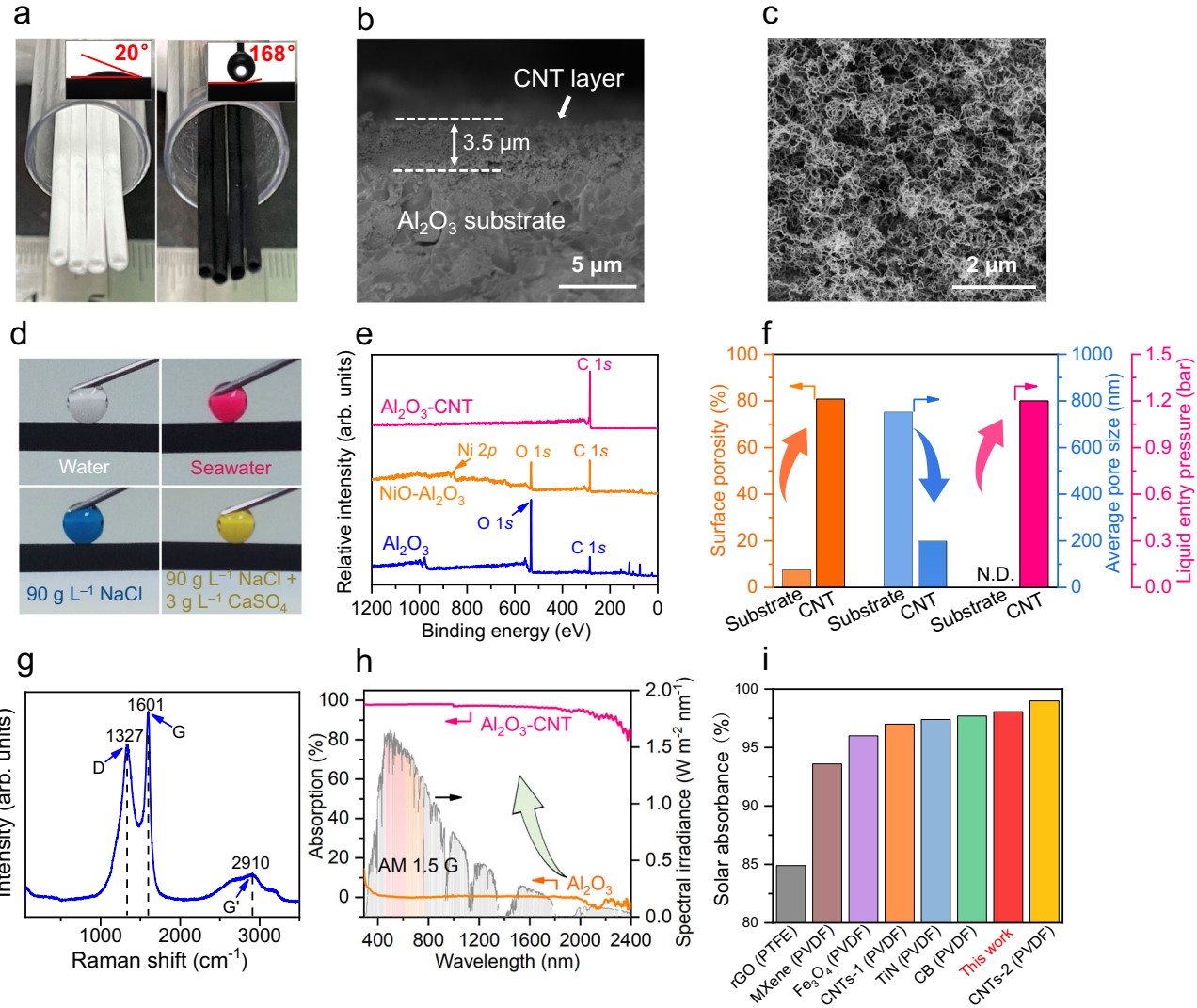

**Fig. 2 | Structure, morphology, and surface properties of ceramic-carbon Janus membranes. a** Photographs of Al$_2$O$_3$ substrate and ceramic-carbon Janus membrane (insets show water contact angles for the substrate and Janus membrane surface). **b** SEM image of membrane cross section with a - 3.5 µm thick solar-thermal CNT network layer. **c** SEM image of the feed-side surface of the ceramic-carbon Janus membrane. **d** Photographs of different solution droplets on the ceramic-carbon Janus membrane surface. **e** Wide-scan XPS spectra of the Al$_2$O$_3$ substrate, NiO-loaded Al$_2$O$_3$ substrate, and ceramic-carbon Janus membrane.

**f** Comparison of surface porosity, average pore size, and liquid entry pressure between Al$_2$O$_3$ substrate and ceramic-carbon Janus membrane (i.e., CNT network), where N.D. denotes not detected. **g** Raman spectrum of the ceramic-carbon Janus membrane. **h** UV-vis−NIR absorption spectra of Al$_2$O$_3$ and ceramic-carbon Janus membranes in the 300−2400 nm wavelength range and standard air mass 1.5 global (AM 1.5 G) solar spectrum. **i** Comparison of solar absorbance between this work and existing state-of-the-art photothermal distillation membranes (Supplementary Table 3).

(3.3 L m$^{-2}$ h$^{-1}$) and high solar-thermal conversion efficiency (66.8%) under simulated solar irradiation (1 kW m$^{-2}$) (Supplementary Fig. 15). A decrease in water flux with increasing salt concentration is observed due to a decreased driving force and increased membrane scaling, which is commonly reported in the literature[6,19].

Elevated feed temperature was also shown to improve desalination performance by enhancing the driving force for water flux across the membrane. In accordance with the Clausius Clapeyron equation and Fick's Law, the increased temperature gradient produces a greater vapor pressure difference across the membrane which accelerates transmembrane water flux (Fig. 4e, Supplementary Fig. 16). Similarly, enhancing solar power density could also improve water flux due to higher interfacial temperature while maintaining stable and high salt rejection. Feed flow rate must also be carefully considered in MD processes. For traditional MD, increasing the flow rate from 2.2 to 6.6 L h$^{-1}$ resulted in a gradual water flux increase from 4.1 ± 0.2 to 4.5 ± 0.3 L m$^{-2}$ h$^{-1}$ (Fig. 4f), because the higher flow rate reduced heat

loss in the direction of feed flow[16]. In contrast, for solar-thermal MD, a slower feed flow rate resulted in higher water flux because it reduced temperature attenuation along the feed-flow direction and thus increased temperature gradient (i.e., driving force) across the membrane. The solar-thermally produced interfacial heat can evidently compensate for tangential heat loss. A feed flow rate of 2.2 L h$^{-1}$ was considered the optimum parameter for the solar-thermal MD process. For constant operation at the optimum feed flow rate, an improved water flux (5.1 ± 0.2 L m$^{-2}$ h$^{-1}$) was observed for room temperature treatment of saline water (35 g L$^{-1}$ NaCl) with simulated solar illumination (1 kW m$^{-2}$) compared to that without solar illumination (4.1 ± 0.1 L m$^{-2}$ h$^{-1}$) (Fig. 4g). In summary, the membrane designed in this work exhibits not only better solar-thermal efficiency (66.8−68.8%), but also higher water flux (3.3−5.1 L m$^{-2}$ h$^{-1}$) than most existing polymeric distillation membranes (Fig. 4h, Supplementary Fig. 17 and Supplementary Table 5) when treating highly saline waters (35−90 g L$^{-1}$).

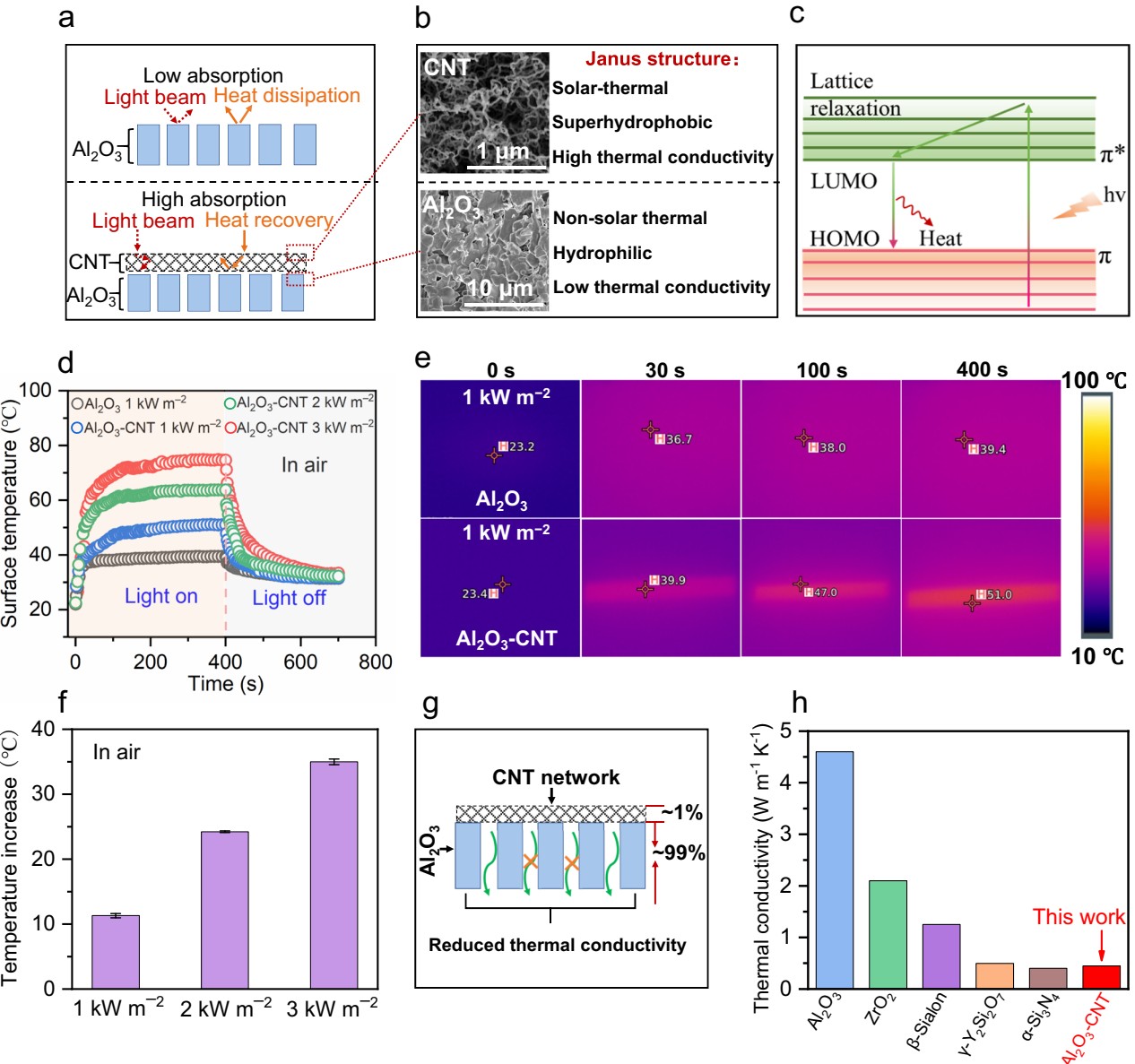

**Fig. 3 | Solar-thermal and thermal-conductive properties of ceramic-carbon Janus membranes. a** Comparison of light and heat capture mechanisms between $Al_2O_3$ substrate and ceramic-carbon Janus membrane. **b** Specially designed Janus structure of ceramic-carbon membrane. **c** Solar-thermal conversion mechanism of CNT network. **d** Surface temperature profiles of $Al_2O_3$ and ceramic-carbon Janus membranes in air under different simulated solar power densities (1–3 kW m⁻²). **e** Infrared thermal images of $Al_2O_3$ and ceramic-carbon Janus membranes before (0 s) and after simulated solar light irradiation (1 kW m⁻²) for different times (30 s, 100 s, 400 s). **f** Temperature increase of ceramic-carbon Janus membranes after illumination for 400 s with different simulated solar power densities (1–3 kW m⁻²) in air. **g** Schematic illustration of the specially designed Janus membrane with low thermal conductivity (the thickness ratio of $Al_2O_3$ to CNT was about 99:1). **h** Comparison of thermal conductivities between reported ceramic membranes ($Al_2O_3$, $ZrO_2$, β-Sialon, γ-$Y_2Si_2O_7$, α-$Si_3N_4$) and the ceramic-carbon Janus membrane designed in this work (Supplementary Table 4). (Error bars represent standard deviation, and data are presented as mean values +/- standard deviation in this study).

## Heat and mass transfer and interfacial evaporation mechanism

To investigate heat and mass transfer mechanisms of conventional and solar-thermal MD, a computational fluid dynamics (CFD) simulation was used which predicts water flux, surface temperature profile, and temperature distribution under different operation conditions (Fig. 5, Supplementary Figs. 18-20). Water flux was simulated at different feed temperatures, salt concentrations, flow rates, and solar illumination intensities, with the simulated results demonstrating good agreement with the experimental results (Fig. 5a, Supplementary Fig. 19). Water flux increased with solar illumination intensity and feed temperature due to the positive exponential relationship between water vapor pressure (i.e., driving force) and temperature (Fig. 5a and Supplementary Fig. 19)[31]. Model validation suggests that the

established modeling method can correctly predict the mass transfer behavior in solar MD processes.

The surface temperature profiles and temperature distributions were also simulated at varying solar power densities (Fig. 5b–e, Supplementary Fig. 20). Without solar illumination (i.e., conventional MD) at a feed temperature of 28 °C, the membrane surface temperature was lower than the bulk feed temperature due to the heat consumption of evaporation (Fig. 5b, c)[32]. However, when the solar power densities were 1 kW m⁻², 2 kW m⁻², and 3 kW m⁻², the membrane surface temperatures (-32.3 °C, 35.3 °C and 37.6 °C) were significantly higher than the bulk feed temperatures in the solar MD process. The increased surface temperatures were due to the solar-thermal effect of the membrane surface where solar light energy was transformed into local

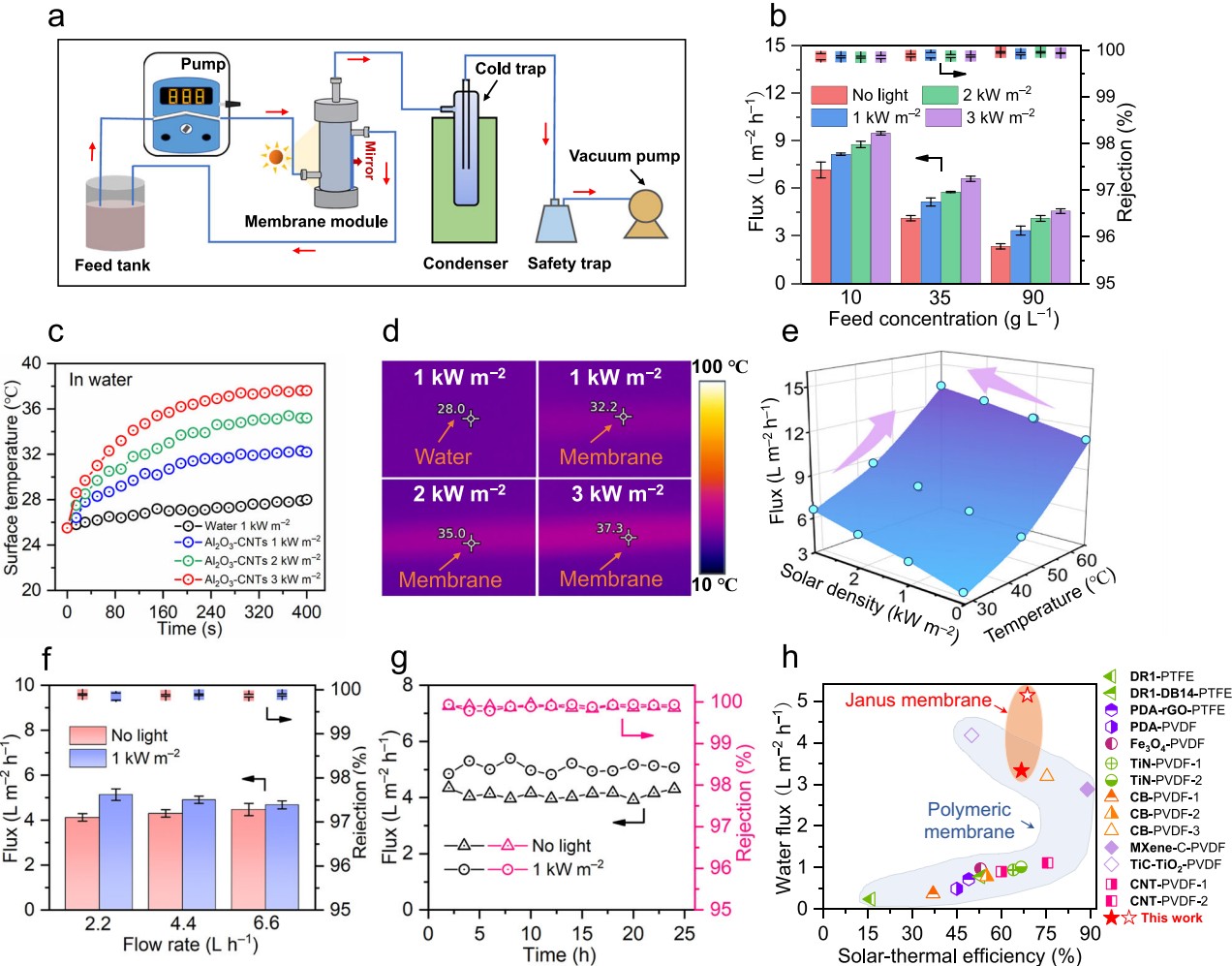

**Fig. 4 | Solar-thermal desalination performance of ceramic-carbon Janus membranes. a** Schematic illustration of a solar-thermally driven desalination setup system. **b** Desalination performance (water flux and salt rejection) of ceramic-carbon Janus membranes for treatment of saline waters with different salt concentrations (10−90 g L⁻¹ NaCl) under simulated solar irradiation with different power densities (0−3 kW m⁻²). **c** Surface temperature profiles of ceramic-carbon Janus membranes in water under different simulated solar power densities (1−3 kW m⁻²). **d** Infrared thermal images of ceramic-carbon Janus membrane surface in water after illumination with different simulated solar power densities (1−3 kW m⁻²) for 400 s. **e** Water flux of ceramic-carbon Janus membranes under different solar power densities and temperatures. **f** Desalination performance of ceramic-carbon Janus membranes at different feed flow rates with and without simulated solar illumination (1 kW m⁻²). **g** Ambient temperature desalination performance (water flux and salt rejection) of ceramic-carbon Janus membrane for treatment of saline waters (35 g L⁻¹ NaCl) with and without simulated solar illumination (1 kW m⁻²). **h** Comparison of water flux and solar-thermal conversion efficiency between existing state-of-the-art solar-thermal MD membranes reported in the literature and the ceramic-carbon Janus membrane designed in this work (empty star: 35 g L⁻¹ NaCl saline solution as feed; solid star: 90 g L⁻¹ NaCl hypersaline solution as feed) (Supplementary Table 5). Error bars represent standard deviation, and data are presented as mean values +/- standard deviation in this study.

heat on the CNT surface. This not only compensated for the heat consumed by evaporation but also further increased membrane interfacial temperature. Similar phenomena could be observed at higher feed operation temperatures of 45 °C and 65 °C (Supplementary Fig. 20). For conventional MD (i.e., without solar illumination), the membrane interface temperature decreased along the feed flow direction (i.e., tangential temperature attenuation in the X direction of Fig. 5b) due to heat loss and exchange. This results in a progressively reduced temperature difference across the membrane (Fig. 5d, e and Supplementary Fig. 20), and consequently a decreased driving force and water flux. In contrast, increased interfacial temperature along the feed flow direction was observed with increasing solar power densities (1−3 kW m⁻²). A temperature polarization coefficient (TPC) was calculated based on experimental and simulation results. Both simulated and experimental TPC values for solar-thermal MD were higher than those of conventional MD (Fig. 5f), indicating an effective mitigation of temperature polarization.

The transport mechanism of water vapor in the Janus membrane was evaluated according to a dusty gas model[9,33] which characterizes mass transfer as occurring through three major mechanisms: Knudsen diffusion, viscous flow, and molecular diffusion. The Knudsen number ($K_n$), defined as the ratio of the mean free path ($\lambda$) and membrane pore size ($d_p$), was used to determine the specific mass transfer regime. Under simulated solar illumination (1 kW m⁻²), the Knudsen number in our study was calculated to be 0.72−0.81 (Fig. 5g, Supplementary Fig. 21). This falls within the 0.01 and 1 range, indicating that a combined mechanism of Knudsen diffusion and viscous flow dominates water vapor transport across the Janus membrane pores (Fig. 5h).

Liquid water on the CNT network membrane surface exists in the Cassie-Baxter state with an ultralow liquid-solid contact fraction (~1.13%) (Supplementary Fig. 22). However, the interfacial water evaporation mechanism remains unclear. A molecular dynamics simulation was performed to investigate the mechanism of evaporation for conventional and solar-thermal MD (Fig. 6, Supplementary Fig. 23).

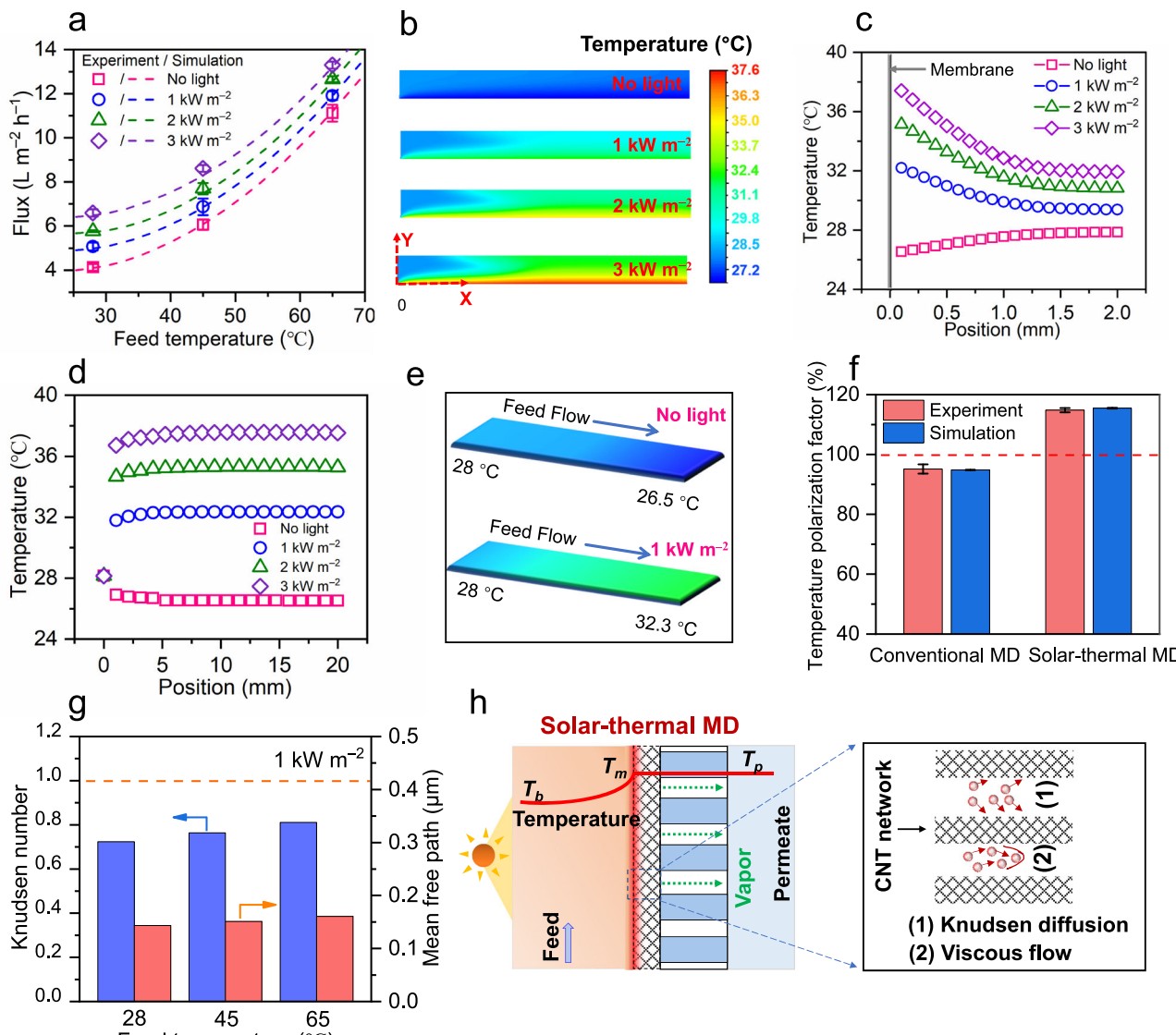

**Fig. 5 | Heat and mass transfer mechanism in conventional and solar-thermal MD processes. a** Experimental and simulated water flux as a function of feed temperature under different solar intensities. **b** Simulated temperature profiles and (**c**) temperature distribution along the direction perpendicular to the membrane surface (i.e., Y direction in (**b**)) under different solar power densities (0–3 kW m⁻²) and a feed temperature of 28 °C. **d** The temperature distribution and (**e**) simulated temperature profiles along the direction parallel to the membrane surface (i.e., X direction in (**b**)) under different light intensities (0–3 kW m⁻²) with a feed temperature of 28 °C. **f** Experimental and simulated temperature polarization factors in conventional MD and solar-thermal MD processes. The dashed line indicates a temperature polarization factor of 100%. **g** Effect of feed temperature on the Knudsen number and mean free path of water vapor molecules under simulated solar illumination (1 kW m⁻²). **h** Codominant mechanisms of enhanced water vapor transport across the Janus membrane: (1) Knudsen diffusion and (2) viscous flow. Error bars represent standard deviation and data are presented as mean values +/- standard deviation in this study.

At 313 K, the evaporative water molecule loss in the water-CNT system (30) is higher than that in the bulk pure water (i.e., water-water) system (13; Fig. 6a and b). Additionally, the water-CNT system lost more water molecules at 353 K (i.e., 46), which was more than twice that of the pure water system (i.e., 22) (Fig. 6c and d). The loss of more water molecules in the water-CNT system can be ascribed to the weakened water-water interactions, which were induced by strong water-CNT interactions. As water molecules interacted with the surface of CNTs, the delocalized π-electron clouds of the CNTs interacted with the water dipoles, facilitating dipole-induced dipole interactions[34–36]. Additionally, although pristine CNTs are hydrophobic and do not directly form hydrogen bonds with water molecules, the presence of CNTs disrupted the hydrogen bond network among water molecules at the surface, leading to the local rearrangement of the water molecular structure[37]. The water distribution images clearly indicate that the introduction of CNTs led to the formation of a looser water layer

structure by disrupting the hydrogen bond network among water molecules (Fig. 6). Furthermore, as the temperature increases, the intensified thermal motion of water molecules weakens their interactions, resulting in an even looser water layer structure. The water-CNT interaction coupled with the increased temperature effectively altered the water state to enable more efficient interfacial evaporation.

To elucidate the mechanism of water state evolution (i.e., the evolution of intermolecular interactions within the water layer), the number of hydrogen bonds and intramolecular energy (i.e., energy barrier for evaporation) in the two systems were investigated during the evaporation process. The number of hydrogen bonds between water molecules on the CNT surface gradually decreased in the 50 picoseconds, whereas in pure water, it exhibited a fluctuating trend at 313 K (Fig. 6e and f). Moreover, the number of hydrogen bonds in the water-CNT system decreased by 61, which was more than twice the decrease of the pure water system (27). At 353 K, the number of lost

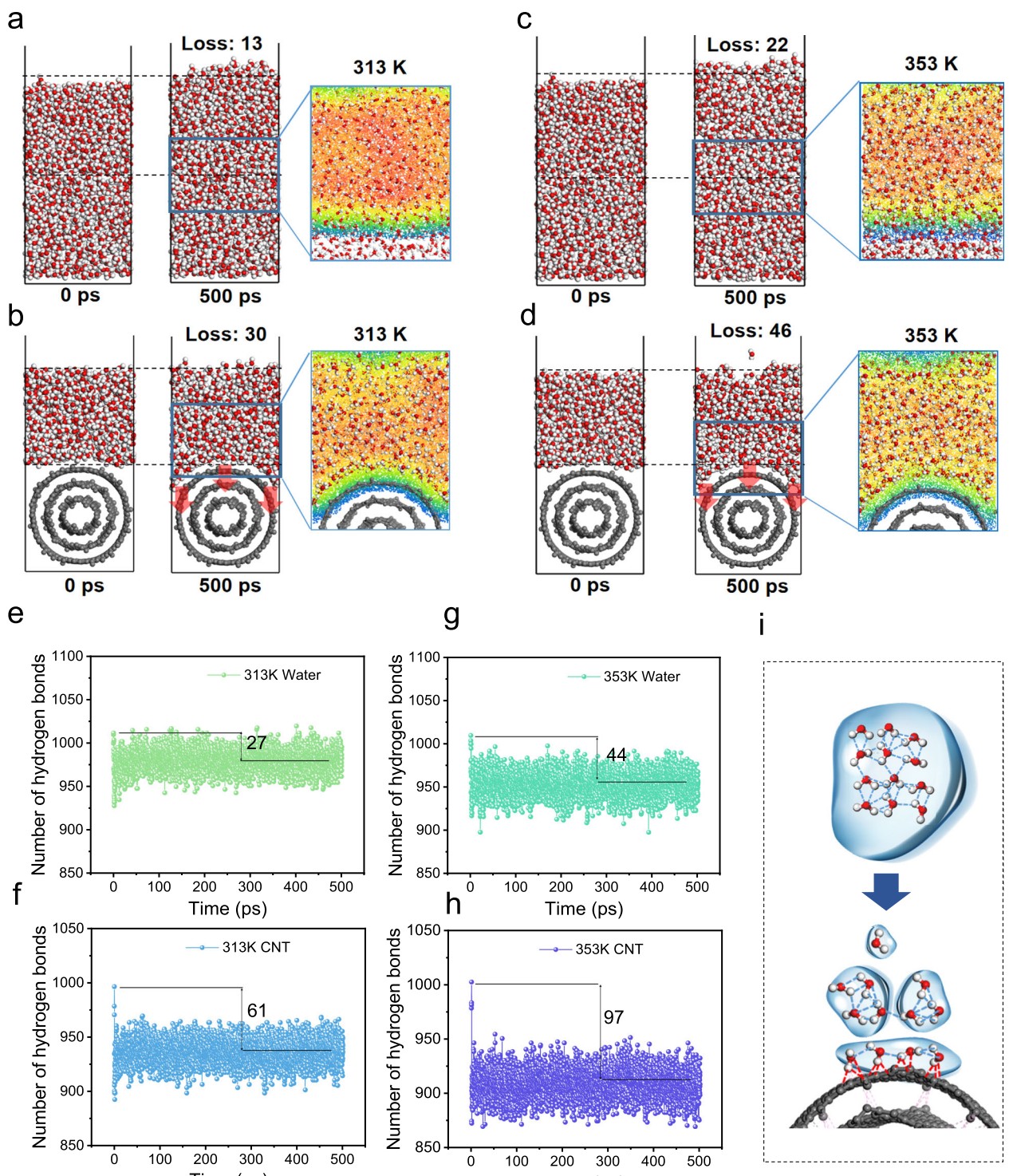

**Fig. 6 | Molecular mechanism of interfacial water evaporation.** Snapshots of the water evaporation process at 313 K and 0 and 500 ps for (**a**) pure water and (**b**) water on the CNT surface. Snapshots of the water evaporation process at 353 K and 0 and 500 ps for (**c**) pure water and (**d**) water on the CNT surface. The enlarged images show the water distribution at the membrane interface. The number of hydrogen bonds between water molecules at 313 K for (**e**) bulk water and (**f**) water

at the CNT surface. The number of hydrogen bonds between water molecules at 353 K for (**g**) bulk water and (**h**) water at the CNT surface. The numbers 27, 61, 44, and 97 indicate the number of lost hydrogen bonds. (**i**) State evolution mechanism of water molecules changing from the bulk state to the interfacial intermediate state (i.e., interacting with the CNT).

hydrogen bonds increased to 93 and 44 for the water-CNT and pure water systems, respectively (Fig. 6g and h). In addition, the water layer of the water-CNT system showed a lower intramolecular energy of 5426.95 kcal mol$^{-1}$ than the pure water system (5633.06 kcal mol$^{-1}$) at 313 K. As the temperature increased to 353 K, the water layer of the

water-CNT system showed a decrease in intramolecular energy (5199.28 kcal mol$^{-1}$), which was still lower than the pure water system (5368.05 kcal mol$^{-1}$) (Supplementary Fig. 23). Generally, water molecules exist as free water (similar to the bulk liquid state), bound water (associated with a material), and intermediate water (between free and

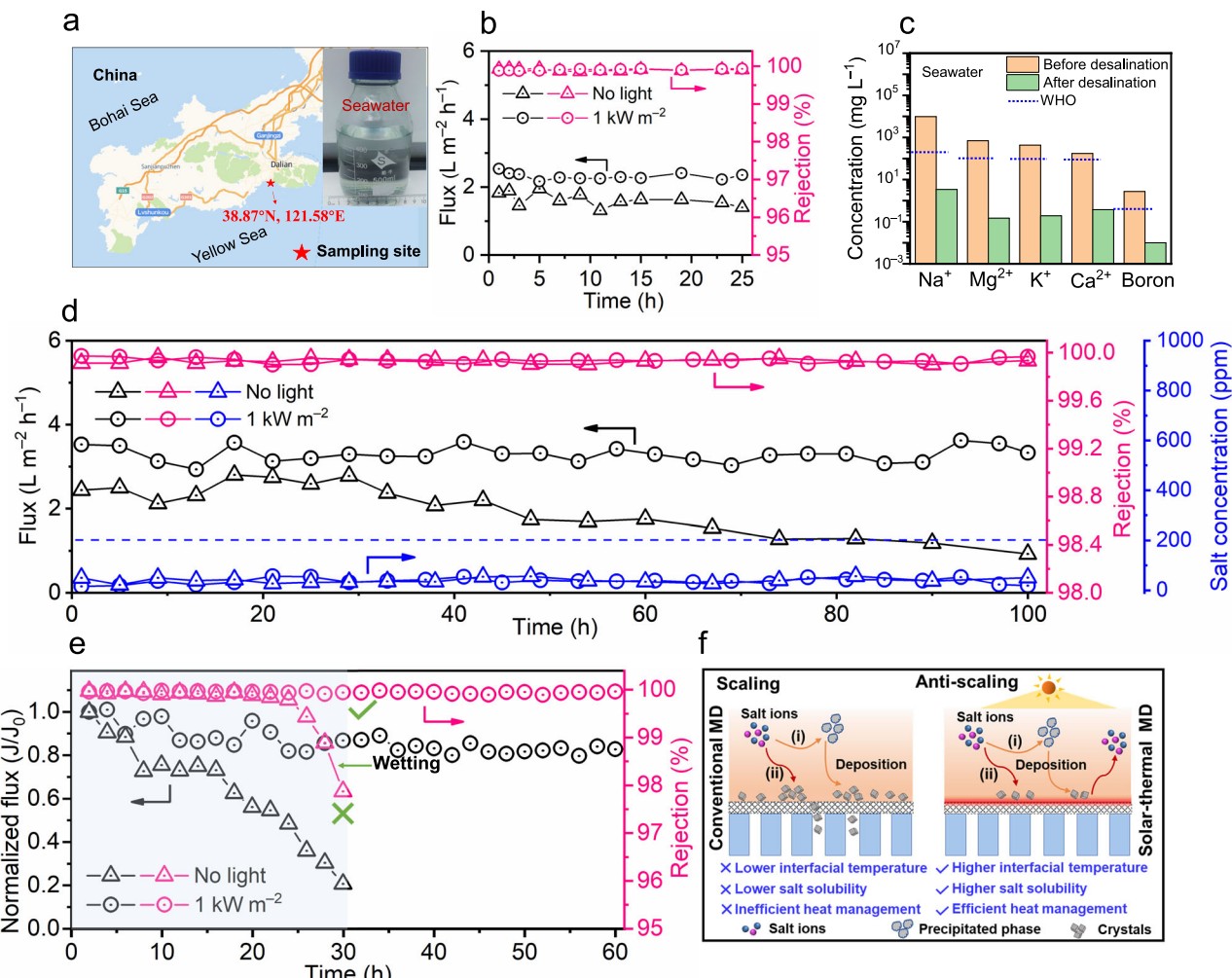

**Fig. 7 | Solar-thermally enhanced desalination performance of ceramic-carbon Janus membranes for the treatment of seawater and hypersaline waters. a** A map of our sampling site where real seawater was collected from the Yellow Sea, China. **b** Ambient temperature desalination performance for treatment of real seawater with and without 1 kW m⁻² solar irradiation. **c** Concentration of five primary species in seawater before and after solar-thermal MD with a ceramic-carbon Janus membrane. **d** Ambient temperature desalination performance for treatment of hypersaline water (90 g L⁻¹ NaCl) for 100 h with and without solar illumination (1 kW m⁻²). **e** Normalized flux and salt rejection of ceramic-carbon Janus membrane for treatment of hypersaline water (90 g L⁻¹ NaCl) containing CaSO₄ (3 g L⁻¹) with and without solar illumination (1 kW m⁻²). **f** Schematic illustration of anti-scaling mechanism of solar-thermal MD (right) outperforming conventional MD (left).

bound states)[38]. Furthermore, intermediate water evaporates much faster than free water[39]. The decreased hydrogen bond numbers and intramolecular energy in this work indicate a transformation of free water molecules into intermediate water molecules in the water-CNT system (Fig. 6i). Since the energy barrier for evaporating intermediate water is lower than that of free water, the CNT layer effectively accelerates water evaporation by inducing the formation of intermediate water.

## Robust desalination for seawater and hypersaline waters

Compared with low-salinity waters, the efficient treatment of real seawater with its complex composition and hypersaline waters presents a greater challenge. The membranes designed in this work show robust operation performance and stability in solar-thermal MD desalination (Fig. 7). Even for ambient temperature treatment of real seawater (Fig. 7a), the membrane exhibits near complete salt rejection (>99.9%) and higher water flux (2.3 ± 0.1 L m⁻² h⁻¹) under solar illumination (1 kW m⁻²) compared to its performance without solar illumination (Fig. 7b). All ion concentrations (Na⁺, Mg²⁺, K⁺, Ca²⁺ and boron) in the permeate were lower than the World Health Organization (WHO) drinking water standard after solar-thermal membrane desalination (Fig. 7c)[40], indicating promising potential for practical application. In

addition, membrane structural morphology was unchanged before and after MD operation (Supplementary Fig. 24). We carried out constant desalination operations for hypersaline water (90 g L⁻¹ NaCl) with and without solar illumination (1 kW m⁻²) at room temperature to further evaluate the membrane operational stability (Fig. 7d). In addition to nearly complete salt rejection (>99.9%), the solar-thermal membrane exhibited higher and more stable water flux (3.1 ± 0.2 L m⁻² h⁻¹) with solar illumination than that without illumination (2.1 ± 0.8 L m⁻² h⁻¹), indicating better performance stability and durability. This can be verified by the SEM-EDS images of the membrane surface which show reduced scaling (i.e., less NaCl accumulation-induced scaling) when operated under solar illumination (Supplementary Fig. 25).

Treating hypersaline water that contains scale-forming minerals such as sulfates is another significant challenge for desalination processes[41]. The formation of mineral scale on the membrane surface usually results in irreversible flux decline and severe pore wetting during MD desalination[42,43]. The Janus membrane in this study exhibited severe flux decline (e.g., ~45% decline at 22 h) when tested without solar illumination, but maintained a high salt rejection (>99.8) for 22 h (Fig. 7e). After 30 h of operation, the salt rejection decreased to ~97.8% as determined by permeate conductivity (~2575.6 μS cm⁻¹)

(Supplementary Fig. 26), indicating the occurrence of scale-induced pore wetting. The salt accumulation-induced scaling is evidenced by a significant change in surface morphology and composition, consistent with NaCl and $CaSO_4$ scale formation (Supplementary Fig. 27). In contrast, under solar illumination, the membrane displays anti-scaling (i.e., anti-salt accumulation) and anti-wetting properties with more stable water flux and nearly complete salt rejection (>99.9%) during long-term operation (Fig. 7e).

The formation of a scaling layer usually involves two pathways: (i) homogeneous nucleation and crystallization in the bulk solution with crystals eventually depositing on the membrane surface, and (ii) heterogeneous nucleation and crystallization on the membrane surface[43,44]. In this study, the conventional MD process suffered from severe mineral scaling, which could be attributed to temperature polarization and transverse temperature attenuation. The solar-thermal MD process, in contrast, demonstrated how elevated membrane surface temperature can mitigate scale formation and wetting. The high interfacial temperature led to enhanced salt solubility at the membrane surface, enabling mineral salts on the CNT interface to re-dissolve into the feed, and reduced temperature polarization and attenuation (Figs. 5, 7f). These results clearly demonstrate that a solar-thermal CNT network layer not only serves as an efficient armor to prevent salt accumulation-induced scaling and surface wetting but also provides a triple-phase membrane interface for rapid water evaporation and efficient vapor transport.

## Discussion

In this work, we rationally designed and fabricated a solar-thermally enhanced Janus membrane for efficient desalination, featuring a triple-phase membrane interface with superhydrophobic and superporous properties and localized solar-thermal conversion. We find that the multi-level pore structure of this membrane provides low thermal conductivity and thereby alleviates temperature polarization and attenuation, which are inherent challenges in conventional MD. For treatment of saline and hypersaline waters, the solar-thermal Janus membrane exhibits higher solar-thermal efficiency (66.8–68.8%) and water flux (3.3–5.1 L m$^{-2}$ h$^{-1}$) than most existing polymeric distillation membranes. CFD simulations indicate that enhanced desalination performance in solar-thermal MD can be ascribed to the increased membrane surface temperature, which mitigates temperature polarization and transverse temperature attenuation. Molecular simulation results reveal that water evaporation is accelerated by the formation of lower H-bonded "intermediate" water molecules on the CNT surface, effectively lowering the energy barrier for interfacial evaporation. Water vapor molecules were found to transport through membrane pores via a combined mechanism of Knudsen diffusion and viscous flow.

The desalination performance of the solar-thermal membrane was tested against conventional MD for real seawater and gypsum-containing hypersaline water, demonstrating superior salt rejection (>99.9%) and stable water flux. The enhanced anti-scaling and wetting resistance of the solar-thermal membranes can be attributed to the stable and elevated membrane surface temperature, which increases salt solubility. The potential treatment applications of these solar-thermal membranes are expected to extend to other important brines and refractory industrial wastewater with nonvolatile components such as various inorganic salts, radioactive ions, and organics. It would also be interesting to systematically investigate the performance and anti-scaling mechanism under various conditions. While the membranes in this work were designed in a tubular configuration with a higher packing density (i.e., higher membrane area per unit volume) than flat-sheet membranes, they may also be fabricated in large flat-sheet configurations for more effective solar-driven interfacial desalination, where surface nanocarbon could more efficiently absorb simulated sunlight (Supplementary Figs. 28, 29, Supplementary

Table 6). Considering future large-scale applications, cost-decreasing or process-simplified strategies need to be employed (such as the use of low-cost membrane materials, and cost-effective catalyst loading methods) since inorganic membranes are more demanding in fabrication or manufacturing costs and conditions. It would be more applicable that the developed membranes could be tested under actual sunlight, which is being investigated as a follow-up to the current work.

## Methods

### Design and fabrication of Janus membrane

The membranes were designed with a dual-layer Janus structure featuring opposite properties on either side of the membrane, such as surface wettability, thermal conductivity, and solar-thermal conversion. $Al_2O_3$ ceramic membranes were prepared by a dry-wet spinning method, which involves phase inversion, drying, and sintering[19]. After optimizing fabrication parameters, the resultant $Al_2O_3$ membranes sintered at 1550 °C with high nitrogen/water permeance and sufficient strength were used as substrates for the ceramic-carbon Janus membranes. Following quantitative control of CNT layer variability and its impact on performance[19], in this work, in situ, construction of the optimized CNT layer on the ceramic substrate was carried out by CVD. To facilitate CNT growth on the ceramic substrate, a nickel oxide (NiO) nano-catalyst (~5 nm) was deposited on the ceramic surface via atomic layer deposition (ALD). The substrate was loaded with catalyst precursor (i.e., nickel oxide) and reduced at 500 °C for 1 h in hydrogen ($H_2$) with a flow rate of 20 mL min$^{-1}$ to form active metallic Ni nano-catalyst. If no NiO layer is deposited, the resulting membranes (i.e., still hydrophilic ceramic membranes) would have poor desalination performance since no CNT layer could be grown without NiO. CNT growth was achieved by introducing a mixture of ethylene (40 mL min$^{-1}$) and hydrogen gases (20 mL min$^{-1}$) in the quartz tube furnace maintaining the reaction at 700 °C for 10 min[6,19]. After the CVD reaction, nitrogen gas (20 mL min$^{-1}$) was used to cool the reactor to ambient temperature (~28 °C).

### Membrane characterization and performance evaluation

Microstructure and surface morphology of the membranes were characterized using field emission scanning electron microscope (FESEM, NOVA NanoSEM 450, American FEI Company, USA). Membrane hydrophilicity/hydrophobicity was measured by a dynamic contact angle goniometer (PT-705, Guangdong Zhongcheng Pussett Equipment Co., Ltd., China). Surface roughness of the membranes was characterized by atomic force microscopy (AFM, Dimension Icon, Bruker, USA). Surface chemical characterization was performed by X-ray photoelectron spectroscopy (XPS, Thermos K-Alpha+, USA). Raman spectra of the membranes were recorded using a Raman spectrometer (inVia Qontor, Renishaw, UK). Absorption spectra were measured in the range of 250–2500 nm in an ultraviolet-visible-near-infrared (UV-vis-NIR) spectrometer (Lambda1050+, PerkinElmer, USA). Thermal conductivity of the membranes was measured using a hot disk thermal constant analyzer (TPS 2500S, Hot Disk AB Company, Sweden). A solar light simulation transmitter (CEL-PE300L-3A, Beijing China Education Au-light Technology Co., Ltd., China) was used to generate the light source. An automatic optical power meter (CEL-NP2000-2(10)A, Beijing China Education Au-light Technology Co., Ltd., China) was used to measure solar power densities. To probe membrane surface temperatures, thermal infrared pictures were randomly captured, which were almost in the middle positions of radial direction of the membranes, by an IR camera with a temperature precision of ±2 °C, measured range of -20~280 °C, IR resolution of 160 × 120 pixel, super IR resolution of 320 × 240 pixel and noise equivalent temperature difference of <0.12 °C (Testo 865, Testo Instrument International Trading Co. Ltd., China). The temperature was calibrated by a thermocouple. A transparent, thin membrane module shell made of quartz

(-1 mm) was used to allow simulated solar light to irradiate the membrane surfaces.

Desalination performance (water flux and salt rejection) of the membranes was systematically investigated in a custom cross-flow filtration mode membrane setup for the treatment of saline waters (10–90 g L$^{-1}$ NaCl), real seawater, and hypersaline water containing gypsum (CaSO$_4$) at different feed temperatures (28–60 °C), feed rate (2.2–6.6 L h$^{-1}$) and solar power densities (0–3 kW m$^{-2}$). Before the experiments were conducted, the membranes were cleaned by tap water to remove residual impurities. Real seawater was obtained from Xinghai Park, China (38.87° N, 121.58° E) and used without any pre-treatment. Supersaturated gypsum solution (CaSO$_4$, 3 g L$^{-1}$) was prepared by mixing NaSO$_4$ and CaCl$_2$ in deionized water. The permeate-side pressure was maintained under near-vacuum (−0.097 MPa) with a vacuum pump and the permeate vapor was condensed in a liquid nitrogen cold trap. Water flux ($J$, L m$^{-2}$ h$^{-1}$) and salt rejection ($R$, %) were calculated using the following equations[2]:

$$J = \frac{\Delta m}{A \cdot \rho \cdot \Delta t} \tag{1}$$

$$R = \left(1 - \frac{C_P}{C_f}\right) \times 100\% \tag{2}$$

where $\Delta m$ (kg) is the mass of the permeate at an interval time $\Delta t$ (h), $A$ (m$^2$) is the effective area of the membrane, and $\rho$ is the density of water (0.9971 g cm$^{-3}$, 25 °C). The salt concentrations (kg L$^{-1}$) of feed ($C_f$) and permeate ($C_p$) were indirectly measured to determine salt rejection by using a conductivity meter (DDS-307A, INESA, Shanghai, China).

Solar-thermal conversion efficiency ($\eta$) was calculated by the following equation[45]:

$$\eta = \frac{(\dot{J} - J)\Delta H}{I} \tag{3}$$

where $\dot{J}$ and $J$ (L m$^{-2}$ h$^{-1}$) is the stable permeate flux with and without solar illumination, respectively, $\Delta H$ is the enthalpy of water vaporization (2441.7 kJ kg$^{-1}$ at 25 °C), and $I$ (kW m$^{-2}$) is the input solar power density.

## CFD simulations
**Computational areas and governing equations.** A two-dimensional CFD model was used to predict and analyze water transport and heat transfer behavior. Calculation areas were modeled based on the membrane module size, as specified in Supplementary Fig. 18. Since the tubular membrane module is symmetrically arranged, only the upper half of the fluid domain was simulated, with rotational symmetry settings applied in Fluent. The model consists of three regions: the feed region, the membrane region, and the permeate region. The dimensions of each region are as follows: feed region of 20 mm × 7.913 mm, membrane region of 20 mm × 0.33 mm, and permeate region of 20 mm × 0.54 mm. The CFD model made the following assumptions: a steady-state condition is achieved and heat loss to the environment is negligible. Governing equations for heat energy, momentum, and mass were considered in the model, which can be expressed as follows[31]:

$$Q = \rho C \mu \nabla T + \nabla q \tag{4}$$

$$\rho(u\nabla)u = -\nabla pI + \nabla \mu(\nabla \mu + (\nabla \mu))^T + F \tag{5}$$

$$\nabla(-B_M \nabla c) = 0 \tag{6}$$

where $Q$ (W m$^{-3}$) is the input heat energy, $\rho$ (kg m$^{-3}$) is the density, $C$ (J kg$^{-1}$ K$^{-1}$) is the heat capacity, $\mu$ (N s m$^{-2}$) is the viscosity, $T$ (K) is the inlet temperature, $q$ (W m$^{-2}$) is the heat flux, u (m s$^{-2}$) is the velocity, $p$ (Pa) is the dynamic pressure, $B_M$ is the mass transfer coefficient (kg m$^{-2}$ s$^{-1}$ Pa$^{-1}$), c (J kg$^{-1}$ K$^{-1}$) is specific heat capacity, $F$ and $I$ are the volume force vector and the unit tensor, respectively. More detailed descriptions about mass and heat transport are presented in 5. Supplementary Methods: Simulation Details and Results.

**CFD mesh and boundary conditions.** SpaceClaim was used to create a computational area of the same size as the membrane module for VMD simulation. ANSYS ICEM was then employed to generate a combined quadrilateral and structured grid over the computational domain. The total number of grids is 18,000 and the local mesh of the fluid domain is given in Supplementary Fig. 18. The inflation function was applied at the membrane-feed interface to refine the mesh, improving the calculation accuracy of the boundary layer. Since the membrane thickness accounts for a small proportion of the fluid domain, the membrane domain does not need to be meshed. Boundary conditions were set to velocity-inlet and outflow-outlet, respectively. At the membrane interface, a porous jump model was used to simulate the permeation process, with consideration of solute retention and solvent flux. The pressure-jump coefficient (C2) of the membrane was calculated based on the membrane porosity and permeability obtained from Darcy's law[46].

$$\Delta p = -\left(\frac{\mu}{\alpha}\nu + C_2 \frac{1}{2}\rho\nu^2\right)\Delta m \tag{7}$$

$$C_2 = \frac{3.5(1-\varepsilon)}{D_p \varepsilon^3} \tag{8}$$

where $p$ (Pa) is the pressure, $\mu$ (N s m$^{-2}$) represents dynamic viscosity, $D_p$ (m) is the average pore diameter, $\varepsilon$ is the porosity, $\alpha$ (m$^2$) is the permeability of the medium, $\nu$ (m s$^{-1}$) is the velocity perpendicular to the membrane pores, $\rho$ (kg m$^{-3}$) is the density of the medium, $\Delta m$ (m) is the medium thickness, and $C_2$ (m$^{-1}$) is the pressure-jump coefficient.

Other physical parameters for boundary conditions for the CFD simulation were obtained from experimental data and numerical equations.

**Slover setting and model calibration.** The governing equations were discretized based on a finite volume approach and solved using ANSYS FLUENT 2020 R2 commercial software. A two-dimensional swirl axisymmetric interface is available in the volume of fluid (VOF) model. Pressure-based solver was used in the steady-state calculations (gravitational acceleration: 9.81 m s$^{-2}$). Standard wall functions and the k-epsilon were set according to near-wall treatment methods and a viscous model. The SIMPLE solution method was applied to a pressure-velocity coupling algorithm. In this procedure, experimental water flux was obtained to check the accuracy of CFD simulation results due to uncertainties in the model input parameters for membrane properties (e.g., tortuosity, porosity, and the size, shape, and distribution of membrane pores). By calibrating the flux, the porous jump membrane surface parameters were adjusted to ensure the simulation results align as closely as possible with experimental results. Then the calibrated boundary parameter was used in the CFD model to predict desalination performance under different operating conditions.

## Molecular dynamics simulations
A molecular dynamics simulation was performed to investigate the state of water during the evaporation process. Model and simulation details are provided in the supporting information (5. Supplementary Methods: Simulation Details and Results).

## Data availability

The data supporting the findings of the study are included in the main text and supplementary information files. Raw data are available from the corresponding authors upon request. Source data are provided with this paper.

## Code availability

All simulation codes are available from the corresponding authors upon request.

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

## Acknowledgements
This work was financially supported by the NSFC-RGC Joint Program (No. 52261160381, Y. Dong) and NSFC General Program (No. 52070033, Y. Dong) of National Natural Science Foundation of China, the General Program of Shenzhen Natural Science Foundation (No. JCYJ20240813113500002, Y. Dong), Pengcheng-Peacock project (Y. Dong), CUHKSZ university development fund (UDF01003350, Y. Dong), and a grant from the Research Grants Council of the Hong Kong Special Administration Region, China (NSFC/RGC Joint Research Scheme N_HKU721/22, C. Tang).

## Author contributions
Y. Dong managed this project. Y. Dong made the concepts and contributions. C. Violet made some important contributions to this paper. Q. Zheng, X. Li and Y. Sun performed the computational simulations. C. Sun performed some experiments. Y. Dong wrote the manuscript. M. Elimelech, Y. Dong, and C. Tang revised the manuscript. All authors approved the manuscript.

## Competing interests
The authors declare no competing interest.
