## [Transparent Peer Review file · Nature Communications]

Ceramic-Carbon Janus Membrane for Robust Solar-Thermal Desalination

Corresponding Author: Professor Menachem Elimelech

Version 0:

Reviewer comments:

Reviewer #1

(Remarks to the Author)

In the current work, the authors report a ceramic-carbon Janus membrane with solar-thermal functionality for enhanced desalination performance, energy efficiency, and stability for hypersaline water treatment. They ascribed enhanced desalination performance to an increased membrane surface temperature, which mitigates temperature polarization and attenuation, thus strengthening the driving force for desalination. This work is interesting, and the reviewer suggests an acceptance after a minor revision.

1. For equations which the authors did not propose, citations are suggested.
2. In the current work, the temperature was measured by the IR camera, how did the authors calibrate the temperature and what's the camera's precision?
3. More details about the CFD simulation are recommended, such as how to treat the porous membrane, the correspondence between the experiment and simulation, and so on.
4. It is obvious that the temperature between the simulation and experiment is different, why? More explanations are suggested.
5. How did the authors consider the salt accumulation in the Janus membrane?
6. In the molecular dynamics simulation part, the authors only consider the interaction between water and the CNT layers. The reasonability needs to be illustrated.
7. More related work in Nature Communications is suggested to be reviewed.

Reviewer #2

(Remarks to the Author)

The ceramic-carbon Janus membrane prepared by this work displayed some MD performances. However, I don't think this work is novel at present. Similar ideas have been published in authors' Nano Letters five years ago. In addition, I don't think this work has great potential for further industrial applications. Are there any advantages compared with polymer membranes? I suggest to reject this work as it cannot meet the standard to be published in NC.

1. How to determine whether the CNT layer obtained under the preparation conditions was optimal for the overall properties of the membrane?
2. In Figure 3h, the alumina materials exhibit relatively high thermal conductivity. Does this mean that changing alumina to zirconia, combined with the phase conversion method, will result in lower thermal conductivity?
3. In Page 9, please supplement the reliability verification of the empirical model.
4. In Figure 4f, the optimal flow rate was selected as 2.2 L h⁻¹. Can the flow rate be further reduced to increase the flux?
5. The characterization methods used to determine the number of hydrogen bonds and intramolecular energy should be described.
6. The explanation for the strong water-CNT interaction is unclear. Please specify what properties of the CNT cause the change in potential energy in water.
7. Is the bond between the CNT layer and the support excellent? Please provide relevant results.
8. If no NiO layer was deposited, will the resulting membrane have poor performance? Please give more results to demonstrate the necessity of this step.

9. The authors claim their Janus membrane design is novel, but fail to clearly differentiate it from existing solar-thermal membrane designs in the literature. Many of the key features like carbon nanotube coatings, ceramic substrates, and solar-thermal effects have been previously reported. The incremental novelty over state-of-the-art is not sufficiently demonstrated.
10. The authors should provide more detailed characterization of the membrane surface, including quantitative analysis of the CNT layer thickness variability and its impact on performance.
11. The potential leaching of carbon nanotubes into the permeate after long-term operation is not addressed, which could have environmental implications.
12. No data on the mechanical strength and long-term durability of the Janus membrane is shown.
13. Deterioration of the photothermal and wetting properties of the CNT layer over time is a concern. Fouling/scaling or oxidation may degrade the solar absorption and hydrophobicity. Longer tests are needed to assess if the surface remains superhydrophobic and the flux remains stable over extended operation in real MD conditions and sunlight exposure.
14. The fabrication method for the Janus membrane is complex, involving multiple steps of spinning, sintering, ALD, and CVD. The scalability and cost of this process for large-scale manufacturing need to be addressed.
15. The enhancements in flux and thermal efficiency are insufficient to establish the practical superiority of this design over state-of-the-art solar-thermal MD. Lack of techno-economic analysis is a major gap.

Reviewer #3

(Remarks to the Author)

The research introduces a novel ceramic-carbon Janus membrane for solar-thermal desalination, showcasing high efficiency and stability. It combines a CNT layer for providing superior hydrophobicity and solar absorption with a ceramic substrate for providing strength, leading to remarkable solar-thermal efficiency (66.8–68.8%) and water flux (3.3–5.1 L m⁻² h⁻¹). The use of advanced simulations such as CFD and MD in this study to elucidate the membrane's performance mechanisms is very interesting and notable. This work significantly advances sustainable water purification, offering a robust solution for treating hypersaline waters, including real seawater and gypsum-containing brines, with potential scalability and broad applicability.

This is a well conducted study about desalination membrane, providing broad and significant interests to the audience of Nature Communications. The membrane design strategy is innovative, and that the proposed structure is consistent with the basic requirements of solar-thermal distillation membranes. I am also impressed by the comprehensive data presented, which well support the critical discussions. Moreover, the achieved efficiency is quite impressive, and I think it sets a new record for solar-thermal distillation membranes.

In summary, I would like to recommend revisions. Some issues need to be explained to address my concerns before acceptance:

1. Despite the advantages of solar-thermal membrane distillation technology in terms of reduced energy consumption, there are shortcomings such as low and unstable water production rates, and therefore this technology may be more suitable for water supply in off-grid or remote areas. Environmental friendliness, affordability of investment and safety of drinking water require particular attention. Inorganic membranes appear to be more demanding in terms of manufacturing costs and conditions. In addition, the carcinogenicity of nickel compounds represented by NiO needs careful attention. It is suggested that the authors could discuss above issues in this manuscript.
2. Line 122, 129, the LEP test fluid (water or others) should be specified.
3. Brines of different concentrations and compositions could provide some practical significance for verifying membrane superhydrophobicity. For membrane superhydrophobicity tests, can the authors use low surface tension liquids such as surfactants or oils? Can the authors test the sliding angle of the brine on the membrane surface?
4. Are the test locations on thermal infrared images randomly selected? The authors should provide some details.
5. What is the filtration mode in this study? Dead-end or cross-flow?
6. What is the difference between two membrane configurations the authors used, such as tube and flat-sheet?
7. The authors should clearly state the derivation of Equation 1 and cite relevant literature.
8. Is the actual seawater pre-treated (microfiltration, etc.) to remove impurities such as particulate matter?
9. The authors sampled real seawater as feed solution, but for solar-thermal membrane distillation, it would be more applicable if the developed membranes could be tested under actual sunlight.
10. It seems that the permeate flux was slightly enhanced after turning on the light. The authors should emphasize the necessity of using this photothermal conversion.
11. The authors provided detailed simulations on mass transfer and heat distribution, and also interesting mechanism analysis on interfacial water evaporation. This provides some useful fundamental scientific insights. The membrane showed good anti-scaling performance. In future, it is recommended to further elaborate the anti-scaling mechanism of the as-developed membranes, especially the effects of their structural features and surface properties (superhydrophobicity, etc.) on the scaling behavior of the membrane surfaces.
12. The section Discussion, should be a brief summary of the most important findings of the study rather than a repetitive description of the contents of the manuscript.
13. The format of references needs to be modified according to the requirement of the journal.

Version 1:

Reviewer comments:

Reviewer #1

(Remarks to the Author)

The response to the comments is appropriate and the reviewer suggests the acceptance of this manuscript.

Reviewer #2

(Remarks to the Author)

It can be accepted now

Reviewer #3

(Remarks to the Author)

The authors have well addressed all my issues. I recommend acceptance of the manuscript in its current form.

Point-to-point Responses to the Reviewers' Comments

Reviewer #1 (Remarks to the Author):

General Comments: In the current work, the authors report a ceramic-carbon Janus membrane with solar-thermal functionality for enhanced desalination performance, energy efficiency, and stability for hypersaline water treatment. They ascribed enhanced desalination performance to an increased membrane surface temperature, which mitigates temperature polarization and attenuation, thus strengthening the driving force for desalination. This work is interesting, and the reviewer suggests an acceptance after a minor revision.

Response: We sincerely appreciate that you took the time to carefully evaluate our manuscript, and that you provided some positive comments on the impact and significance of our work, “this work is interesting, and the reviewer suggests an acceptance after a minor revision”. We also appreciate your recommendation of accepting after a minor revision. We have carefully considered the comments raised by you, rethought some statements and made revisions, and provided detailed point-to-point responses.

Your positive comments on our manuscript were instrumental for us to reflect on the framing and impact of our work.

1. For equations which the authors did not propose, citations are suggested.

Response: We agree with your opinion that equations should be cited. To address your concerns, after deleting Equation-1 (original manuscript), we have further carefully checked and cited Equation-1 (Equation-2, original manuscript), Equation-2 (Equation-3, original manuscript) (please see Page 23), Equation-7 (Equation-8, original manuscript) and Equation-8 (Equation-9, original manuscript) (please see Page 24) while Equation-3, Equation-4, Equation-5, Equation-6 (Equations 4-7, original manuscript) have been cited when submitted.

2. In the current work, the temperature was measured by the IR camera, how did the authors calibrate the temperature and what's the camera's precision?

Response: Thank you for this valuable suggestion. To address your concerns, we have made further revisions by providing more details of the IR camera such as precision (please see Page 22, revised manuscript), which are shown as follows.

To probe membrane surface temperatures, thermal infrared pictures were randomly captured, which were almost in the middle positions of radial direction of the membranes, by an IR camera with a temperature precision of $\pm 2^{\circ}\text{C}$, measured range of $-20 - 280^{\circ}\text{C}$, IR resolution of 160×120 pixel, super IR resolution of 320×240 pixel and noise equivalent temperature difference of $< 0.12^{\circ}\text{C}$ (Testo 865, Testo Instrument International Trading Co. Ltd., China). The temperature was calibrated by a thermocouple at factory.

3. More details about the CFD simulation are recommended, such as how the treat the porous membrane, the correspondence between the experiment and simulation, and so on.

Response: Thank you for this valuable suggestion. To address your concerns, we have made further revisions by providing more details about the CFD simulation (please see Page 23, Page 24, revised manuscript), which are shown as follows.

Since the tubular membrane module is symmetrically arranged, only the upper half of the fluid domain was simulated, with rotational symmetry settings applied in Fluent. The model consists of three regions: the feed region, the membrane region, and the permeate region. The dimensions of each region are as follows: feed region of 20 mm × 7.913 mm, membrane region of 20 mm × 0.33 mm, and permeate region of 20 mm × 0.54 mm.

The inflation function was applied at the membrane-feed interface to refine the mesh, improving the calculation accuracy of boundary layer.

At the membrane interface, a porous jump model was used to simulate the permeation process, with consideration of solute retention and solvent flux. The pressure-jump coefficient (C_2) of the membrane was calculated based on the membrane porosity and permeability obtained from Darcy's law.⁴⁶

$$\Delta p = -\left(\frac{\mu}{\alpha}v + C_2 \frac{1}{2}\rho v^2\right)\Delta m \quad (8)$$

$$C_2 = \frac{3.5(1-\varepsilon)}{D_p \varepsilon^3} \quad (9)$$

where μ represents dynamic viscosity, D_p is the average pore diameter, ε is the porosity, α is the permeability of the medium, v is the velocity perpendicular to the membrane pores, ρ is the density of the medium, Δm is the medium thickness, and C_2 is the pressure-jump coefficient.

By calibrating the flux, the porous jump membrane surface parameters were adjusted to ensure the simulation results align as closely as possible with experimental results.

Reference

46. Bassel A. Abdelkader, Mostafa H. Sharqawy, Pressure drop across membrane spacer-filled channels using porous media characteristics and computational fluid dynamics simulation, *Desalination and Water Treatment*, 247 (2022) 28–39

4. It is obvious that the temperature between the simulation and experiment is different, why? More explanations are suggested.

Response: Thank you for this question. After careful check, we would like to clarify that the simulation temperature (40°C, 313 K) (Fig. 6) is close to the experiment temperature (37.3 °C) (Fig.4). Considering the effect of higher feed temperature, we also studied the number of hydrogen bonds and intermolecular interaction energy at higher simulation temperature (80°C, 353 K).

5. How did the authors consider the salt accumulation in the Janus membrane?

Response: Thank you for this question. Salt accumulation is indeed an important practical application issue for the operation of Janus membranes. In this work, we indeed considered the fouling and scaling (i.e., salt accumulation) of the Janus membranes during constant operation treatment of seawater and hypersaline waters (Fig. 7, Figs. S24, S25, S26, S27). We find that the membranes show mitigated scaling (i.e., less NaCl accumulation-induced scaling) for treatment of hypersaline waters ($90 \text{ g L}^{-1} \text{ NaCl}$) when operated under solar illumination. We further find that the membranes show mitigated scaling (i.e., less NaCl and CaSO_4 accumulation-induced scaling) for treatment of gypsum-containing hypersaline waters ($90 \text{ g L}^{-1} \text{ NaCl}$ and $3 \text{ g L}^{-1} \text{ CaSO}_4$) when operated under solar illumination.

To address your concerns, we have made detailed discussion with further minor revisions (please see Page 17, Page 18), which are shown as follows.

Compared with low-salinity waters, efficient treatment of composition-complex real seawater and hypersaline waters is more challenging. The membranes designed in this work show robust operation performance and stability in solar-thermal MD desalination (Figure 7).

We carried out constant desalination operations for hypersaline water ($90 \text{ g L}^{-1} \text{ NaCl}$) with and without solar illumination (1 kW m^{-2}) at room temperature to further evaluate the membrane operational stability (Figure 7d). In addition to nearly complete salt rejection ($>99.9\%$), the solar-thermal membrane exhibited higher and more stable water flux ($3.1 \pm 0.2 \text{ L m}^{-2} \text{ h}^{-1}$) with solar illumination than that ($2.1 \pm 0.8 \text{ L m}^{-2} \text{ h}^{-1}$) without illumination, indicating better performance stability and durability. This can be verified by the SEM-EDS images of the membrane surface which show mitigated scaling (i.e., less NaCl accumulation-induced scaling) when operated under solar illumination (Figure S25).

The Janus membrane in this study exhibited severe flux decline (e.g., $\sim 45\%$ decline at 22 hours) when tested without solar illumination, but maintained a high salt rejection (>99.8) for the first 22 hours (Figure 7e). After 30 hours of operation, the salt rejection significantly decreased to $\sim 97.8\%$ as determined by permeate conductivity ($\sim 2575.6 \mu\text{S cm}^{-1}$) (Figure S26), indicating the occurrence of scale-induced pore wetting. The salt accumulation induced scaling is evidenced by a significant change in surface morphology and composition, consistent with NaCl and CaSO_4 scale formation (Figure S27). In contrast, under solar illumination, the membrane displays anti-scaling (i.e., anti salt accumulation) and anti-wetting properties with more stable water flux and nearly complete salt rejection ($>99.9\%$) during long-term operation (Figure 7e). The formation of a scaling layer usually involves two pathways: (i) homogeneous nucleation and crystallization in the bulk solution with crystals eventually depositing on membrane surface, and (ii) heterogeneous nucleation and crystallization on the membrane surface.^{39,40} In this study, the conventional MD process suffered from severe mineral scaling, which could be attributed to temperature polarization and transverse temperature attenuation. The solar-thermal MD process, in contrast, demonstrated how elevated membrane surface temperature can mitigate scale formation and wetting. The high interfacial temperature led to enhanced salt solubility at the membrane surface, enabling mineral salts on the CNT interface to re-dissolve into the feed, and reduced temperature polarization and attenuation (Figure 5, Figure 7f). These results clearly demonstrate that a

solar-thermal CNT network layer not only serves as an efficient armor to prevent salt accumulation induced scaling and surface wetting, but also provides a unique triple-phase membrane interface for rapid water evaporation and efficient vapor transport.

6. In the molecular dynamics simulation part, the authors only consider the interaction between water and the CNT layers. The reasonability needs to be illustrated.

Response: Thank you for this valuable suggestion. We would like to clarify that we indeed only considered the interaction between water molecules and the CNT layers, as well as water-water intermolecular interaction because the ceramic substrate of our Janus membrane did not interact with water molecules due to the superhydrophobic feature of CNT layer surface (Please see Fig. S22).

To address your concerns, we have made further minor revisions (please see Page S31, revised supporting information), which are shown as follows.

In addition to water-water intermolecular interaction, the interaction between water molecules and CNT layers was only considered because the ceramic substrate of our Janus membrane did not interact with water molecules due to the superhydrophobic feature of CNT layer surface.

7. More related work in Nature Communications is suggested to be reviewed.

Response: Thank you for this valuable suggestion. We have reviewed some works and cited three relevant papers from the journal Nature Communications. We would like to clarify that the citation only depends on the relevance of the work.

Reviewer #2 (Remarks to the Author):

General Comments: The ceramic-carbon Janus membrane prepared by this work displayed some MD performances. However, I don't think this work is novel at present. Similar ideas have published in authors' Nano Letters five years ago. In addition, I don't think this work has great potential for further industrial applications. Are there any advantages compared with polymer membranes? I suggest to reject this work as it cannot meet the standard to be published in NC.

Response: We sincerely appreciate you for providing detailed comments and suggestions on how to improve the manuscript. We would like to clarify the key difference from the former paper, the novel aspects and contributions, industrial application potential and advantages over polymer membranes, which are shown with details as follows.

I. The key difference from the former paper: We would like to clarify the difference between our published paper (*Nano Letters* 18 (9) (2018) 5514 – 5521.) and the current work. The structure and properties of ceramic-carbon Janus membranes was systematically optimized to obtain the best overall properties and performance in our former work (*Nano Letters* 18 (9) (2018) 5514 – 5521.). Different from the former work focusing on structural optimization and conventional membrane distillation performance (*Nano Letters* 18 (9) (2018) 5514 – 5521.), the key motivation of the current work is to investigate heat and mass transfer and interfacial evaporation mechanisms, as well as the desalination performance and stability of more challenging real seawater, hypersaline or gypsum-containing hypersaline brines via novel solar-thermal membrane distillation process.

II. Important novel aspects and contributions are described below:

To address your concerns about novelty, we have further highlighted the key contributions and novel aspects of our work and its broader implications, which motivated our decision to submit to *Nature Communications*.

1. The first novel aspect in this work is **the proposed concept Janus membrane for solar-thermal desalination** (Fig. 3). Most reports employed the design and fabrication methods such as incorporation or coating of solar-thermal materials into or onto conventional distillation membranes (Table S5), but they usually reduce surface porosity, decrease vapor permeability and thus water treatment efficiency. Unlike them, we employed a different strategy of *in situ* growth of superhydrophobic CNT onto robust ceramic membranes to form a Janus membrane structure, with different properties such as surface wettability, conductivity, and solar-thermal conversion. As Reviewer #3 positively commented in the this round of review, “**The membrane design strategy is innovative, and that the proposed structure is consistent with the basic requirements of solar-thermal distillation membranes**”. More importantly, as fully demonstrated in our work, such a special design enables enhanced performance and operation stability, which is still challenging for conventional distillation membranes.

2. The novel second aspect of this work is the **superior performance and heat/mass transfer mechanism of our Janus solar-thermal membrane** in comparison to both

conventional MD and other state-of-the-art solar-thermal distillation membranes. We experimentally demonstrate that this membrane exhibits higher solar-thermal efficiency and water flux than most existing polymeric solar-thermal distillation membranes when treating saline and hypersaline waters (Fig. 4h, Table S5). This endows the membranes with higher water treatment efficiency, and consequently greater application potential. We wish to emphasize that, in addition to experimental data, we employed **advanced computational fluid dynamics simulation** protocols. We find that enhanced desalination performance can be ascribed to an increased membrane surface temperature, which mitigates temperature polarization and attenuation, thus enhancing the driving force for desalination.

3. The novel third aspect is the characterization of the molecular mechanism for enhanced interfacial water evaporation induced by specially designed nano-carbon membrane surface (Fig. 6). While published works on solar-thermal distillation membranes largely focus on membrane fabrication and performance evaluation, very rarely is the fundamental scientific mechanism for interfacial water evaporation, which remains unclear. To address such a challenging issue, we, once again, employed **advanced molecular dynamics simulation** in our work. Interestingly, we found that the nano-carbon membrane surface accelerates water evaporation by inducing a water state conversion (i.e., from free water molecules to intermediate water molecules) at the membrane interface with decreased hydrogen bonding and a lower evaporation energy barrier (i.e., intramolecular energy) (Fig. 6). Moreover, we also find that water vapor molecules transport through the membrane pores by a combined mechanism of Knudsen diffusion and viscous flow (Fig. 5g, h). We strongly believe this fundamental understanding has much broader scientific implications and will aid in bridging knowledge gaps across different disciplines and fields of science, technology, and engineering (e.g., chemistry, material science, environmental science, chemical engineering, water and membrane technologies). Thus, we believe that this work with fundamental mechanistic insights is of great interest for the broad readership of *Nature Communications*.

4. The fourth novel aspect of this work is the enhanced stability under practical or complex conditions (i.e., real seawater, hypersaline water containing CaSO_4) of our Janus membrane (Fig. 7, Fig. S24-27). Differing from previous reports focusing on desalination under mild feed conditions (synthetic saline water $35 \text{ g L}^{-1} \text{ NaCl}$, Tables S5), we intentionally subjected our Janus membrane to harsh or practical conditions for prolonged periods of time to evaluate their resilience in practical applications that have proven challenging for conventional distillation membranes. To this end, **our membrane showcased stable performance when treating real seawater or hypersaline water**, both of which have complicated compositions, which remain challenging for conventional MD desalination. Stable operation under complex conditions is of great practical significance for membrane technologies. We believe that this has broader implications for the rational design of inorganic distillation membranes, which has not been the traditional focus in the field. Our current work provides an entirely new platform for robust desalination under harsh conditions, expanding the applications where membrane-based desalination is now relevant.

III. Industrial application potential and advantages over polymer membranes

We agree with your opinion that industrial application potential is important for membranes. However, we strongly believe that immediate scaling potential of membrane fabrication process is only a minor element for a high-impactful scientific

paper. Fabrication of a ceramic-carbon Janus membrane is experimentally challenging. Despite this, we successfully fabricated a high-quality ceramic-carbon Janus membrane for solar-thermal desalination applications. In fact, if we directly use commercial ceramic membranes as substrate, our fabrication process is relatively straightforward, only requiring two steps: (i) deposition of nano-catalysts and (ii) in situ growth of CNT. Furthermore, we have demonstrated that it is technically feasible to employ large-size ceramic membranes (15cm×15cm) for Janus membrane fabrication (Figs. S28, S29, Table S4). Moreover, the core novel aspects of this work include the proposed Janus membrane concept, molecular mechanism of enhanced water evaporation, superior performance and heat/mass transfer mechanism, and harsh operation stability, all of which have not been explored in previous works. Additionally, the stable performance of treating composition-complex real seawater and hypersaline brines (Fig. 7, Figs. S24, S25, S26, S27, Table S4) indicates the promising practical potential of our membranes.

IV. Advantages over polymer membranes

We would like to clarify that in our original manuscript, we have demonstrated that the Janus membrane designed in this work exhibits stronger solar absorption than most polymeric membranes (Fig. 2i, Table S3), and importantly higher solar-thermal efficiency (66.8–68.8%) and water flux (3.3–5.1 L m⁻² h⁻¹) than most existing polymeric solar-thermal distillation membranes when treating saline and hypersaline waters (Fig. 4h, Fig. S17, Table S3). We also believe that the Janus membrane would outperform some polymeric membranes in thermal, mechanical and chemical stability due to the highly robust feature of ceramic and nano-carbon materials.

Moreover, in this round of review, Reviewer #1, Reviewer #3 have both provided positive and detailed comments on the novelty and impact of our work. Reviewer #1 recognized our work with quite positive comments, “this work is interesting, and the reviewer suggests an acceptance after a minor revision”. Particularly, Reviewer #3 recognized our work with very positive comments, some of which are also shown as follows.

“The research introduces a novel ceramic-carbon Janus membrane for solar-thermal desalination, showcasing high efficiency and stability.”

“The use of advanced simulations such as CFD and MD in this study to elucidate the membrane's performance mechanisms is very interesting and notable. This work significantly advances sustainable water purification, offering a robust solution for treating hypersaline waters, including real seawater and gypsum-containing brines, with potential scalability and broad applicability.”

“This is a well conducted study about desalination membrane, providing broad and significant interests to the audience of Nature Communications. The membrane design strategy is innovative, and that the proposed structure is consistent with the basic requirements of solar-thermal distillation membranes. I am also impressed by the comprehensive data presented, which well support the critical discussions. Moreover, the achieved efficiency is quite impressive, and I think it sets a new record for solar-thermal distillation membranes.”

1. How to determine whether the CNT layer obtained under the preparation conditions was optimal for the overall properties of the membrane?

Response: Thank you for this helpful question. The structure and properties of ceramic-carbon Janus membranes was systematically optimized to obtain the best overall properties and performance in our former work (*Nano Letters* 18 (9) (2018) 5514 – 5521.). Different from the former work focusing on structural optimization and conventional membrane distillation performance (*Nano Letters* 18 (9) (2018) 5514 – 5521.), the key motivation of the current work is to investigate heat and mass transfer and interfacial evaporation mechanisms, as well as the desalination performance and stability of more challenging real seawater, hypersaline or gypsum-containing hypersaline brines via novel solar-thermal membrane distillation process.

To address your concerns, we have made further revisions by citing our former paper (please see Page 4, revised manuscript), which are shown as follows.

Different from the former work focusing on structural optimization and conventional membrane distillation performance¹⁹, the key motivation of the current work is to investigate heat and mass transfer and interfacial evaporation mechanisms, as well as the desalination performance and stability of more challenging real seawater and hypersaline brines via solar-thermal membrane distillation process.

2. In Figure 3h, the alumina materials exhibit relatively high thermal conductivity. Does this mean that changing alumina to zirconia, combined with the phase conversion method, will result in lower thermal conductivity?

Response: Yes! We believe that lower thermal conductivity could be realized if we just only change membrane materials from alumina ($\sim 4.600 \text{ Wm}^{-1}\text{K}^{-1}$ for homogeneous membrane pore structure) to zirconia ($\sim 2.100 \text{ Wm}^{-1}\text{K}^{-1}$ for homogeneous membrane pore structure) while keeping the same fabrication processes (i.e., phase conversion, sintering and chemical vapor deposition etc.) and consequently similar composite membrane structure (Table S4). We are sorry that we intend to do such experiments in the follow up to this work because they are far from our current concerns and we wish to spend our time on more important topics. We hope you can kindly understand our situations and decision.

Nevertheless, we have provided full explanation with more details on this issue (please see Page 8, revised manuscript), which are shown as follows.

This is attributed to the low thermal conductivity of the specially designed Al_2O_3 substrate, which comprises multi-level pore structures with long finger-like macro-voids (Figure S5). The Janus membrane exhibits similar thermal conductivity as the substrate because the Al_2O_3 to CNT thickness ratio is $\sim 99:1$. Due to such a specially designed substrate structure with higher porosity, the Janus membrane shows lower thermal conductivity than conventional Al_2O_3 ceramic membrane with a homogeneous particulate-packing pore structure (i.e., lower porosity) (Figure S5).³⁰ Low thermal conductivity could be expected if only substrate materials are changed (e.g., from alumina to zirconia) while keeping the same fabrication processes and consequently similar composite membrane structure.

3. In Page 9, please supplement the reliability verification of the empirical model.

Response: Thank you for your suggestion. We must apologize that after careful discussion, we have decided to delete Equation 1 since we think that we have provided sufficient discussion on the effect of temperature and solar power density on

water flux (Page 10, revised manuscript), which is also shown as follows.

Elevated feed temperature was also shown to improve desalination performance by enhancing the temperature gradient across the membrane and thereby accelerating water flux due to the improved water vapor pressure difference and thus driving force based on the Clausius Clapeyron equations and Fick's Law (Figure 4e, Figure S16). Similarly, enhancing solar power density could also improve water flux due to higher interfacial temperature while maintaining stable and high salt rejection.

4. In Figure 4f, the optimal flow rate was selected as 2.2 L h⁻¹. Can the flow rate be further reduced to increase the flux?

Response: We believe that theoretically higher water flux would be expected if we further reduce flow rate. But during our experiments, the flow rate fluctuated more significantly when flow rate was reduced lower than 2.2 L h⁻¹. The fluctuation of flow rate would also result in the instability of water flux. This is why the optimal flow rate was selected as 2.2 L h⁻¹.

5. The characterization methods used to determine the number of hydrogen bonds and intramolecular energy should be described.

Response: Thank you for valuable suggestions. We agree with your opinion that the detailed characterization methods of determining the number of hydrogen bonds and intramolecular energy should be described. To address your concerns, we have made further revisions (please see Page S31-32, revised Supporting Information), which are shown as follows.

The intramolecular energy was determined using the Cohesive Energy Density Analysis Tool in Materials Studio using COMPASS III Force field, which incorporates bond stretching, angle bending, torsional interactions, and non-bonded interactions. COMPASS III is one of the universal force fields based on ab initio simulations. The force field for water in COMPASS III is a flexible model incorporating the inter- and intramolecular motion of a water molecule, which is appropriate for investigating the formation of water molecules microscopically.²¹ The cutoff radius of short-range interactions was set to 1.25 nm, and the long-range electrostatic interactions were calculated by the Ewald method.²² The charge distribution in a water molecule was set to -0.41 eC for hydrogen and 0.82 eC for oxygen, which were determined using COMPASS III. The carbon atoms constituting MWCNT are charge-neutral.

The hydrogen bond analysis was performed using the Hydrogen Bond Analysis Tool in Materials Studio. A hydrogen bond is considered to exist when the distance between the donor and acceptor is less than 2.5 Å, and the donor-hydrogen-acceptor angle is greater than 90°.

Reference

21. Maekawa, Y.; Sasaoka, K.; Yamamoto, T. Structure of water clusters on graphene: A classical molecular dynamics approach. *Jpn. J. Appl. Phys.* 2018, 57 (3), 035102. <https://doi.org/10.7567/JJAP.57.035102>.

22. Ewald, P. P. Die Berechnung optischer und elektrostatischer Gitterpotentiale. *Ann. Phys.* 1921, 369

6. The explanation for the strong water-CNT interaction is unclear. Please specify what properties of the CNT cause the change in potential energy in water.

Response: Thank you for valuable suggestions. We agree with your opinion that the explanation on strong water-CNT interaction should be clear by specifying the effect of CNT on potential energy in water. To address your concerns, we have made further revisions for better clarification (please see Page 15, revised manuscript), which are shown as follows.

The loss of more water molecules in the water-CNT system can be ascribed to the weakened water-water interactions. As water molecules interacted with the surface of CNTs, the delocalized π -electron cloud of the CNTs interacted with the water dipole, facilitating dipole-induced dipole interactions.³⁴⁻³⁶ Additionally, although pristine CNTs are hydrophobic and do not directly form hydrogen bonds with water molecules, the presence of CNTs disrupted the hydrogen bond network among water molecules at the surface.³⁷ leading to the local rearrangements of water molecular structure. The water distribution images clearly indicate that the introduction of CNTs led to the formation of a looser water layer structure by disrupting the hydrogen bond network among water molecules (Figure 6). Furthermore, as the temperature increases, the intensified thermal motion of water molecules weakened their interactions, resulting in an even looser water layer structure.

To confirm the evolution mechanism of water states (i.e., interaction evolution within water layer), the number of hydrogen bonds and intramolecular energy (i.e., energy barrier for evaporation) in the two systems were investigated during the evaporation process.

Reference

34. Alexiadis, A.; Kassinos, S. Molecular Simulation of Water in Carbon Nanotubes. *Chem. Rev.* 2008, 108 (12), 5014-5034. <https://doi.org/10.1021/cr078140f>.

35. Lu, D.; Li, Y.; Ravaioli, U.; Schulten, K. Empirical Nanotube Model for Biological Applications. *J. Phys. Chem. B* 2005, 109 (23), 11461-11467. <https://doi.org/10.1021/jp050420g>.

36. Arab, M.; Picaud, F.; Devel, M.; Ramseyer, C.; Girardet, C. Molecular selectivity due to adsorption properties in nanotubes. *Phys Rev B* 2004, 69 (16), 165401. <https://doi.org/10.1103/PhysRevB.69.165401>.

37. Walther, J. H.; Jaffe, R.; Halicioglu, T.; Koumoutsakos, P. Carbon Nanotubes in Water: Structural Characteristics and Energetics. *J. Phys. Chem. B* 2001, 105 (41), 9980-9987. <https://doi.org/10.1021/jp011344u>.

7. Is the bond between the CNT layer and the support excellent? Please provide relevant result.

Response: We well understand your concerns that the bond between the CNT layer and the support should be important for constant membrane operation applications.

We have done the high shear force cross-flow filtration experiments at different flow rates (0 to $6.28 \text{ m}\cdot\text{s}^{-1}$) in our former work (*Nano Letters* 18 (9) (2018) 5514 – 5521.).

After simple cleaning for impurities removal (60 mins), there is no further mass loss, even when exposed to a high flow rate of $6.28 \text{ m}\cdot\text{s}^{-1}$. This suggests there is good adhesion between the ceramic substrate and CNTs, endowing the Janus membrane with excellent structural stability. This can assure the stable operation of the solar-thermal MD processes in the present work. Before the performance tests were conducted, the membranes were first cleaned by tap water to remove some impurities. It is worth mentioning that solar membrane distillation is a stationary evaporation process, not a pressure-driven filtration process, and the detachment of membrane components is also strongly limited in absence of hydrostatic pressure.

Figure S18. Effect of water flow time on residual mass percent of the ceramic-CNT composite membranes cleaned with tap water at different cross-flow rates in the range of 0 to $6.28 \text{ m}\cdot\text{s}^{-1}$. (*Nano Letters* 18 (9) (2018) 5514 – 5521.).

8. If no NiO layer was deposited, will the resulting membrane have poor performance? Please give more results to demonstrate the necessity of this step.

Response: Thank you for valuable comments and suggestions. Yes! If no NiO layer is deposited, the resulting membranes (i.e., still hydrophilic ceramic membranes) will have no desalination performance since no CNT layer could be grown on the ceramic substrate surface (we have already provided Fig. S11 in the supporting information file). After reduction, NiO layer became nano-sized metallic Ni, which acted as the catalyst for *in situ* growth of CNT network in our work. We have already studied the role of NiO in the formation mechanism of CNT in our previous papers (1, 2). Nevertheless, to address your concerns, we have provided some more explanation on this issue (please see Page 21, revised manuscript), which are shown as follows.

To facilitate CNT growth on the ceramic substrate, a nickel oxide (NiO) nano-catalyst ($\sim 5 \text{ nm}$) was first deposited on the ceramic surface via atomic layer deposition (ALD). The substrate was loaded with catalyst precursor (i.e., nickel oxide) and reduced at $500 \text{ }^\circ\text{C}$ for 1 h in hydrogen (H_2) with a flow rate of 20 mL min^{-1} to form active metallic Ni nano-catalyst. If no NiO layer is deposited, the resulting membranes (i.e., still hydrophilic porous ceramic membranes) would have no desalination performance

since no CNT layer could be grown on the ceramic substrate surface. CNT growth was achieved by introducing a mixture of ethylene (40 mL min^{-1}) and hydrogen gases (20 mL min^{-1}) in the quartz tube furnace maintaining the reaction at $700 \text{ }^\circ\text{C}$ for 10 min .^{6, 19}

Figure S11. (a, b) Surface SEM images and (c, d) Image J processed images of (a, c) ceramic membrane (without NiO deposition) and (b, d) ceramic-carbon Janus membranes (with NiO deposition). The blue area in Figures 11c and 11d denotes surface porosity.

References

6. Yiran Si, Chunyi Sun, Dongfeng Li, Fenglin Yang, Chuyang Y. Tang, Xie Quan, Yingchao Dong, Michael D. Guiver, Flexible superhydrophobic metal-based carbon nanotube membrane for electrochemically enhanced water treatment, *Environmental Science & Technology*, 54 (14) (2020) 9074–9082.
19. Yingchao Dong, Lining Ma, Chuyang Y. Tang, Fenglin Yang, Xie Quan, David Jassby, Michael J. Zaworotko, Michael D. Guiver, Stable superhydrophobic ceramic-carbon-nanotube composite desalination membranes, *Nano Letters* 18 (9) (2018) 5514–5521.

9. The authors claim their Janus membrane design is novel, but fail to clearly differentiate it from existing solar-thermal membrane designs in the literature. Many of the key features like carbon nanotube coatings, ceramic substrates, and solar-thermal effects have been previously reported. The incremental novelty over state-of-the-art is not sufficiently demonstrated.

Response: Thank you for valuable comments and suggestions. One of the novel

aspects in this work is the proposed concept Janus membrane for solar-thermal desalination (Fig. 3). Most reports employed the design and fabrication methods such as incorporation or coating of solar-thermal materials into or onto conventional distillation membranes (Table S5), but they usually reduce surface porosity, decrease vapor permeability and thus water treatment efficiency. Unlike them, we employed a different strategy of in situ growth of superhydrophobic CNT network onto robust ceramic membranes to form a dual-layer Janus membrane structure, with different properties such as surface wettability, conductivity, and solar-thermal conversion. The key novelties in our Janus membrane design are special Janus structure, superporous and superhydrophobic surface, endowing enhanced solar-thermal conversion, high flux and permeability, and promising desalination performance and stability.

As Reviewer #3 positively commented in the this round of review, “The membrane design strategy is innovative, and that the proposed structure is consistent with the basic requirements of solar-thermal distillation membranes”. More importantly, as fully demonstrated in our work, such a special design enables enhanced performance and operation stability for efficiently treating more challenging hypersaline waters and seawater, which is still challenging for conventional distillation membranes.

To address your concerns, we have provided some explanation on this issue (please see Page 4, revised manuscript), which are shown as follows.

Herein, we report a Janus solar-thermal ceramic-carbon membrane with a triple-phase membrane interface for efficiently treating more challenging hypersaline waters and seawater. Unlike conventional distillation membranes, the key novelties in our Janus membrane design are special Janus structure, superporous and superhydrophobic surface, endowing enhanced solar-thermal conversion, high flux and permeability, and promising desalination performance and stability.

10. The authors should provide more detailed characterization of the membrane surface, including quantitative analysis of the CNT layer thickness variability and its impact on performance.

Response: Thank you for valuable comments and suggestions. We do agree with your opinion that it is important to characterize membrane surface, including quantitative analysis of CNT layer variability and its impact on performance.

In our previous paper (*Nano Letters* 18 (9) (2018) 5514 – 5521), we have already studied the effect of different CNT layers on the properties and performance of the resultant membranes. Please see Figure 1, Figure 2, as well as Figure S14, Figure S15, Figure S16, Figure S20, Figure S23 (supporting information). We have provided quantitative analysis of CNT layer variability (Figure S14, Figure S15, Figure S16, shown as follows) and its impact on the properties and performance of the membranes (Figure 1, Figure 2, Figure S20, Figure S23 etc.).

Figure S14. (a) CNT yield variation with Ni(NO₃)₂ concentrations for Ni nano-catalyst loading (CVD reaction temperature 650 °C, CVD reaction time 3 h), typical surface SEM images of the ceramic-CNT composite membranes fabricated using different Ni(NO₃)₂ concentrations: (b) 10 wt.%, (c) 20 wt.% and (d) 30 wt.%. (Nano Letters 18 (9) (2018) 5514 - 5521)

Figure S15. (a) CNT yield variation with CVD reaction temperature (nickel nitrate concentration 30 wt. % for Ni nano-catalyst loading, reaction time 3 h). Typical SEM surface images of the ceramic-CNT composite membranes fabricated at different CVD reaction temperatures for 3 h: (b) 550 °C, (c) 650 °C and (d) 750 °C. (Nano Letters 18 (9) (2018) 5514 - 5521)

Figure S16. (a) Progression of CNT yield at different CVD reaction times (nickel nitrate concentration 30 wt.% for Ni nano-catalyst loading, CVD reaction temperature 650 °C), and typical surface SEM images of the ceramic-CNT composite membranes fabricated at 650 °C for different CVD reaction times (b) 45 min, (c) 120 min and (d) 180 min. The red solid circle in Fig. S16a indicates membrane crumbling at a long CVD reaction time of 210 min. The insert in Fig. S16d shows an enlarged surface SEM image of the ceramic-CNT membrane fabricated at a CVD reaction time of 180 min, having superporous membrane surface structure with randomly grown CNT networks. (*Nano Letters* 18 (9) (2018) 5514 – 5521)

Nevertheless, to address your concerns, we have provided some more explanation on this issue (please see Line Page 21, revised manuscript), which are shown as follows.

Following quantitative control of CNT layer variability and its impact on performance¹⁹, in this work, in situ construction of the optimized CNT layer on the ceramic substrate was carried out by chemical vapor deposition (CVD).

References

19. Yingchao Dong, Lining Ma, Chuyang Y. Tang, Fenglin Yang, Xie Quan, David Jassby, Michael J. Zawortko, Michael D. Guiver, Stable superhydrophobic ceramic-carbon-nanotube composite desalination membranes, *Nano Letters* 18 (9) (2018) 5514–5521.

11. The potential leaching of carbon nanotubes into the permeate after long-term operation is not addressed, which could have environmental implications.

Response: Thank you for valuable comments and suggestions. We do agree with your opinion that the potential leaching of carbon nanotubes and other impurities could have some environmental impact. To address this issue, we have done some pre-treatment for all the membranes before performance test in this work. We also have done leaching experiments in the former work (*Nano Letters* 18 (9) (2018) 5514 – 5521.).

Before the experiments were conducted, the membranes were first cleaned by tap

water to remove some minor impurities. After simple cleaning for impurities removal (60 mins), there is no further mass loss, even when exposed to a high flow rate of $6.28 \text{ m}\cdot\text{s}^{-1}$ (please see the figure and the response to the Comment-7). This suggests there is good adhesion between the ceramic substrate and CNTs, endowing the Janus membrane with excellent structural stability, stable operation and no leaching during the solar-thermal MD processes in the present work. It is worth mentioning that solar membrane distillation is a stationary evaporation process, not a pressure-driven filtration process, and the detachment of membrane components is also strongly limited in absence of hydrostatic pressure.

Regarding the potential environmental risk of heavy metal compounds in the substrates, we have also already considered well this issue in our former paper (Section “S3.8 Safety assessment of FC-CNT membranes for DCMD application”, *Nano Letters* 18 (9) (2018) 5514 – 5521.). The leaching concentration of Ni, Al, Fe, Mn was much lower than those of drinking water criterion (WHO). This can assure the safety of the solar-thermal MD processes in the present work.

To address your concerns, we have provided some revisions (please see Page 22, revised manuscript), which are shown as follows.

Before the performance tests were conducted, the membranes were first cleaned by tap water to remove residual impurities.

12. No data on the mechanical strength and long-term durability of the Janus membrane is shown.

Response: Thank you for valuable comments and suggestions. We do agree with your opinion that the mechanical strength and long-term durability of the Janus membranes are important. The mechanical strength of the Janus membranes only depends on the substrates since the ratio of substrate to CNT is as high as 99:1 (Fig. 3). In our original submission, we have provided the mechanical strength (Figures S4e, S4f). It is worth mentioning that solar membrane distillation is a stationary evaporation process, not a pressure-driven filtration process, and the long-term strength durability of the membranes keep well in absence of hydrostatic pressure.

Figure S4. Properties of aluminum ceramic membranes sintered at different temperature: (a) pore size distribution, (b) average pore size, (c) N₂ permeance, (d) water permeance, (e) typical load-deflection curves and (f) bending strength.

13. Deterioration of the photothermal and wetting properties of the CNT layer over time is a concern. Fouling/scaling or oxidation may degrade the solar absorption and hydrophobicity. Longer tests are needed to assess if the surface remains superhydrophobic and the flux remains stable over extended operation in real MD conditions and sunlight exposure.

Response: Thank you for valuable comments and suggestions. We appreciate your concern regarding the performance stability of the membranes in our study. We also agree with your opinion that the deterioration of photothermal and wetting properties of the CNT layer over time is an important concern and that longer tests are important over extended operation, considering the possible fouling, scaling or oxidation of the CNT layer. To address this issue, we have provided long-term operation performance data for the treatment of challenging hypersaline brines in our original submission (Figure 7, Figs. S24, S25, S26, S27, Table S4). Actually, the operation duration in our work outperforms most reports only except for three studies (see following Table). In most studies, there are no flux and rejection performance over operation time, or limited flux and rejection performance over short-term operation time.

In our work, we found that compared with the membranes without illumination, the solar-thermal membrane exhibited higher and more stable water flux under solar illumination, indicating better performance stability and durability for the treatment of hypersaline water (Figure 7d). Under solar illumination, the membrane displays anti-scaling (i.e., anti salt accumulation) and anti-wetting properties with more stable water flux and nearly complete salt rejection (>99.9%) for the treatment of hypersaline water containing scale-forming minerals such as sulfates during long-term operation (Figure 7e). These results indicate minor or limited fouling, scaling or oxidation during operation, and also limited degradation of surface solar absorption and hydrophobicity. The stable performance of treating composition-complex real seawater and hypersaline brines (Fig. 7, Figs. S24, S25, S26, S27, Table S4) indicates the promising practical potential of our membranes.

More importantly, we would like to clarify that our work primarily focuses on the following core novel aspects, including the proposed Janus membrane concept, molecular mechanism of enhanced water evaporation, superior performance and heat/mass transfer mechanism, and harsh operation stability, all of which have not been explored in previous works. In order to not defocus on these novel aspects, we would like to perform longer-term stable performance test in our followup work. We hope you could understand our situation and decision.

Table Comparison of operation duration between Janus membrane designed in this work and reported solar-thermal desalination membranes.

Membrane	Photothermal material	Feed composition	Operation Duration/h	Refs.
PVDF-2	CNT	35 g L ⁻¹ NaCl	/	8
PVDF	Fe ₃ O ₄	35 g L ⁻¹ NaCl	8	4
PVDF-1	TiN	35 g L ⁻¹ NaCl	240	13
PVDF-1	CNT	35 g L ⁻¹ NaCl	/	14
PVDF-2	TiN	35 g L ⁻¹ NaCl	48	6
PTFE	DR1	35 g L ⁻¹ NaCl	/	15
PTFE	DR1-DB14	Artificial seawater*	/	16
PVDF-2	CB	Seawater	48	7
PVDF-1	CB	10 g L ⁻¹ NaCl	/	17
PVDF	PDA	29 g L ⁻¹ NaCl	/	18
PVDF-3	CB	35 g L ⁻¹ NaCl	/	19
C-PVDF	MXene	100 g L ⁻¹ NaCl	120	3
PVDF	TiC-TiO ₂	30 g L ⁻¹ NaCl	144	20
PTFE	PDA-rGO	40 g L ⁻¹ NaCl	/	2
Janus	CNT	35 g L ⁻¹ NaCl	100	This work

References

- (3) Zhang, B.; Wong, P. W.; Guo, J.; Zhou, Y.; Wang, Y.; Sun, J.; Jiang, M.; Wang, Z.; An, A. K. Transforming Ti₃C₂T_x MXene's intrinsic hydrophilicity into superhydrophobicity for efficient photothermal membrane desalination. *Nat. Commun.* **2022**, *13*, (1), 1-10.
- (4) Li, W.; Chen, Y.; Yao, L.; Ren, X.; Li, Y.; Deng, L. Fe₃O₄/PVDF-HFP photothermal membrane with in-situ heating for sustainable, stable and efficient pilot-scale solar-driven membrane distillation. *Desalination* **2020**, *478*, 114288.
- (6) Farid, M. U.; Kharraz, J. A.; An, A. K. Plasmonic titanium nitride nano-enabled membranes with high structural stability for efficient photothermal desalination. *ACS Applied Materials & Interfaces* **2021**, *13*, (3), 3805-3815.
- (7) Gong, B.; Yang, H.; Wu, S.; Yan, J.; Cen, K.; Bo, Z.; Ostrikov, K. K. Superstructure-enabled anti-fouling membrane for efficient photothermal distillation. *ACS Sustain. Chem. Eng.* **2019**, *7*, (24), 20151-20158.
- (8) Han, X.; Wang, W.; Zuo, K.; Chen, L.; Yuan, L.; Liang, J.; Li, Q.; Ajayan, P. M.; Zhao, Y.; Lou, J. Bio-derived ultrathin membrane for solar driven water purification. *Nano Energy* **2019**, *60*, 567-575.
- (13) Zhang, Y.; Li, K.; Liu, L.; Wang, K.; Xiang, J.; Hou, D.; Wang, J. Titanium nitride

nanoparticle embedded membrane for photothermal membrane distillation. *Chemosphere* **2020**, *256*, 127053.

(14) Huang, J.; Hu, Y.; Bai, Y.; He, Y.; Zhu, J. Novel solar membrane distillation enabled by a PDMS/CNT/PVDF membrane with localized heating. *Desalination* **2020**, *489*, 114529.

(15) Fujiwara, M. Water desalination using visible light by disperse red 1 modified PTFE membrane. *Desalination* **2017**, *404*, 79-86.

(16) Fujiwara, M.; Kikuchi, M. Solar desalination of seawater using double-dye-modified PTFE membrane. *Water Res.* **2017**, *127*, 96-103.

(19) Chen, Y.-R.; Xin, R.; Huang, X.; Zuo, K.; Tung, K.-L.; Li, Q. Wetting-resistant photothermal nanocomposite membranes for direct solar membrane distillation. *J. Membr. Sci.* **2021**, *620*, 118913.

(20) Liu, J.; Guo, H.; Sun, Z.; Li, B. Preparation of photothermal membrane for vacuum membrane distillation with excellent anti-fouling ability through surface spraying. *J. Membr. Sci.* **2021**, *634*, 119434.

14. The fabrication method for the Janus membrane is complex, involving multiple steps of spinning, sintering, ALD, and CVD. The scalability and cost of this process for large-scale manufacturing need to be addressed.

Response: Thank you for valuable comments and suggestions. We agree with your opinion that the scalability and cost of the process for large-scale manufacturing is important for membranes in industrial applications. However, we strongly believe that immediate scaling potential of membrane fabrication process is only a minor element for a high-impactful scientific paper. Fabrication of a ceramic-carbon Janus membrane is challenging. Despite this, we successfully fabricated a high-quality ceramic-carbon Janus membrane for solar-thermal desalination applications. In fact, if we directly use large-size commercial ceramic membranes as substrate, our fabrication process is relatively straightforward, only requiring two steps: (i) deposition of nano-catalysts by ALD and (ii) in situ growth of CNT by CVD. Furthermore, we have demonstrated that it is technically feasible to employ **large-size ceramic membranes** (15cm×15cm) for Janus membrane fabrication (Figs. S28, S29, Table S4). Moreover, the core novel aspects of this work include the proposed Janus membrane concept, molecular mechanism of enhanced water evaporation, superior performance and heat/mass transfer mechanism, and harsh operation stability, all of which have not been explored in previous works. Additionally, the stable performance of treating composition-complex real seawater and hypersaline brines (Fig. 7, Figs. S24, S25, S26, S27, Table S4) indicates the promising practical potential of our membranes.

To address your concerns, we have made further revisions (please see Page 20, revised manuscript), which are shown as follows.

While the membranes in this work were designed in a tubular configuration with a higher packing density (i.e., higher membrane area per unit volume) than flat-sheet membranes, they may also be fabricated in large-size flat-sheet configurations for more effective solar-driven interfacial desalination, where surface nanocarbon could more efficiently adsorb simulated sunlight (Figure S28, S29, Table S6). Considering future large-scale application, some cost-decreasing or process-simplified strategies

need to be employed (such as the use of low-cost membrane materials, and cost-effective catalyst loading methods) since inorganic membranes are more demanding in fabrication or manufacturing costs and conditions.

15. The enhancements in flux and thermal efficiency are insufficient to establish the practical superiority of this design over state-of-the-art solar-thermal MD. Lack of techno-economic analysis is a major gap.

Response: Thank you for valuable comments and suggestions. We do agree with your opinion that the techno-economic analysis of desalination technology is important for potential practical applications.

To address your concerns, we have made techno-economic analysis for our Janus membrane and state-of-the-art solar-thermal polymeric distillation membranes (Figure S17b, Table S5), and made further revisions (please see Page S23, revised supporting information), which are shown as follows.

The techno-economic analysis of desalination technique is important for potential practical applications. To this end, the cost-effectiveness of membrane desalination was defined and then calculated, which reflects the water flux performance per unit mass membrane cost with a unit of $\text{L m}^{-2} \text{h}^{-1} \text{USD}^{-1}$. This indicator can represent the economics of desalination techniques more accurately. Compared to the state-of-the-art solar-thermal polymeric distillation membranes, our Janus ceramic-carbon membranes exhibit not only higher water flux, but also a higher cost-effectiveness when treating brines with similar salt concentration (Figure S18, Table S5). In addition, the cost can be further reduced in large-scale massive production of membranes. Both inexpensive membrane materials and reproducible fabrication processes of this Janus membrane would be positively beneficial to future large-scale application and commercialization.

Figure S17. Comparison of (a) Δ flux (i.e., the difference of water flux with and without solar illumination) and solar-thermal conversion efficiency, (b) cost-effectiveness between existing state-of-the-art solar-thermal polymeric distillation membranes reported in the literature and the ceramic-carbon Janus membrane (i.e., CNT-Ceramic) fabricated in this work (solid pentagram, line-filled bar: 90 g L^{-1} NaCl, hollow pentagram, bar: 35 g L^{-1} NaCl) (Table S5).

Table S5. Comparison of solar-thermal desalination performance between ceramic-carbon Janus membrane designed in this work and reported solar-thermal desalination membranes.

Membrane	Photothermal material	Feed	Temperature (°C)	Photothermal efficiency (%)	Solar density (kW m ⁻²)	Δ Flux (L m ⁻² h ⁻¹)	Flux (L m ⁻² h ⁻¹)	Cost-effectiveness (L m ⁻² h ⁻¹ USD ⁻¹)	Rejection (%)	Refs.
PVDF-2	CNT	35 g L ⁻¹ NaCl	N.A.	75.6	1	1.11	1.11	45.15	99.8	8
PVDF	Fe ₃ O ₄	35 g L ⁻¹ NaCl	25	53.0	1	0.81	0.97	42.29	99.9	4
PVDF-1	TiN	35 g L ⁻¹ NaCl	20	64.1	1	0.94	0.94	40.86	99.6	13
PVDF-1	CNT	35 g L ⁻¹ NaCl	25	60.0	1	0.90	0.90	36.61	99.9	14
PVDF-2	TiN	35 g L ⁻¹ NaCl	23.3	66.7	1	1.01	1.01	43.90	99.9	6
PTFE	PDA-rGO	40 g L ⁻¹ NaCl	N.A.	49.0	1	0.72	0.72	26.72	99.9	2
PTFE	DR1	35 g L ⁻¹ NaCl	N.A.	16.0	1	0.24	0.24	10.64	99.8	15
PTFE	DR1-DB14	Artificial seawater	26.6	53.0	1	0.78	0.78	34.58	99.9	16
PVDF-1	CB	10 g L ⁻¹ NaCl	25	~37.0	0.7	~0.37	~0.37	16.13	99.5	17
PVDF	PDA	29 g L ⁻¹ NaCl	20	45.0	0.75	0.49	0.49	8.00	99.9	18
PVDF-2	CB	Seawater	20	~55.1	1	~0.78	0.78	34.00	99.6	7
PVDF-3	CB	35 g L ⁻¹ NaCl	35	75.4	1	1.13	3.19	139.07	99.9	19
PVDF	TiC-TiO ₂	30 g L ⁻¹ NaCl	30	~50.0	1	~0.74	~4.18	175.44	99.95	20
Janus	CNT	35 g L ⁻¹ NaCl	28	68.8	1	1.02	5.14	203.79	99.9	This work
C-PVDF	MXene	100 g L ⁻¹ NaCl	30	89.0	1	1.33	2.88	53.86	99.9	3
Janus	CNT	90 g L ⁻¹ NaCl	28	66.8	1	0.99	3.33	132.03	99.9	This work

Notes: PVDF: polyvinylidene fluoride, PTFE: polytetrafluoroethylene, CNT: carbon nanotube, Fe₃O₄: ferroferric oxide, TiN: titanium nitride, PDA-rGO:

polydopamine-reduced graphene oxide, DR1: disperse red 1, DR1-DB14: disperse red 1-disperse blue 14, CB: carbon black. Artificial seawater were prepared from NaCl, CaCl₂·2H₂O, KCl, KBr, SrCl₂, LiCl, MnCl₂·4H₂O, AlCl₃·6H₂O and Na₂WO₄·6H₂O. N.A.: not available

Reviewer #3 (Remarks to the Author):

General Comments: The research introduces a novel ceramic-carbon Janus membrane for solar-thermal desalination, showcasing high efficiency and stability. It combines a CNT layer for providing superior hydrophobicity and solar absorption with a ceramic substrate for providing strength, leading to remarkable solar-thermal efficiency (66.8–68.8%) and water flux (3.3–5.1 L m⁻² h⁻¹). The use of advanced simulations such as CFD and MD in this study to elucidate the membrane's performance mechanisms is very interesting and notable. This work significantly advances sustainable water purification, offering a robust solution for treating hypersaline waters, including real seawater and gypsum-containing brines, with potential scalability and broad applicability.

This is a well conducted study about desalination membrane, providing broad and significant interests to the audience of Nature Communications. The membrane design strategy is innovative, and that the proposed structure is consistent with the basic requirements of solar-thermal distillation membranes. I am also impressed by the comprehensive data presented, which well support the critical discussions. Moreover, the achieved efficiency is quite impressive, and I think it sets a new record for solar-thermal distillation membranes.

In summary, I would like to recommend revisions. Some issues need to be explained to address my concerns before acceptance:

Response: We sincerely appreciate that you took the time to carefully evaluate our manuscript, and that you provided very positive comments on the impact and significance of our work. We also appreciate your recommendation of accepting after revisions. We have carefully considered the comments raised by you, rethought and revised some statements, made detailed revisions and provided detailed point-to-point responses.

Your positive comments on our manuscript were instrumental for us to reflect on the framing and impact of our work.

1. Despite the advantages of solar-thermal membrane distillation technology in terms of reduced energy consumption, there are shortcomings such as low and unstable water production rates, and therefore this technology may be more suitable for water supply in off-grid or remote areas. Environmental friendliness, affordability of investment and safety of drinking water require particular attention. Inorganic membranes appear to be more demanding in terms of manufacturing costs and conditions. In addition, the carcinogenicity of nickel compounds represented by NiO needs careful attention. It is suggested that the authors could discuss above issues in this manuscript.

Response: Thank you for valuable comments and suggestions. We do agree with your opinion that solar-thermal membrane distillation technology may be more suitable for water supply in off-grid or remote areas where environmental friendliness, affordability of investment and safety of drinking water are highly required. Also, we agree that inorganic membranes are more demanding in fabrication or manufacturing costs and conditions. Regarding the potential environmental risk of nickel compounds, we have also already considered well this issue in our former paper (Section “S3.8 Safety assessment of FC-CNT membranes for DCMD application”, *Nano Letters* 18 (9) (2018) 5514 – 5521.). The leaching concentration of Ni was lower than the

detection limit (i.e., ~ 0), which is lower than that ($0.07 \mu\text{g}\cdot\text{L}^{-1}$) of drinking water criterion (WHO). This can assure the safety of the solar-thermal MD processes in the present work.

To address your concerns, we have further made some revisions (please see Page 3, Page 20, revised manuscript), which are show as follows.

To address these issues, solar-thermal distillation membranes were proposed, which enable water supply in off-grid or remote areas where environmental friendliness, affordability of investment and safety of drinking water are required. Solar-thermal distillation membranes combine conventional hydrophobic distillation membranes with solar-thermal materials that convert renewable solar energy into localized thermal energy on the membrane surface.

Considering future large-scale application, some cost-decreasing or process-simplified strategies need to be employed since inorganic membranes are more demanding in fabrication or manufacturing costs and conditions.

2. Line 122, 129, the LEP test fluid (water or others) should be specified.

Response: Thank you for this suggestion. The LEP test fluid was water. To address your concerns, we have further specified the LEP test fluid by making minor revisions “since their water liquid entry pressure (LEP)”, “higher water liquid entry pressure (1.2 bar)”. (please see Page 6, revised manuscript).

3. Brines of different concentrations and compositions could provide some practical significance for verifying membrane superhydrophobicity. For membrane superhydrophobicity tests, can the authors use low surface tension liquids such as surfactants or oils? Can the authors test the sliding angle of the brine on the membrane surface?

Response: Thank you for these comments! We agree with your opinion that the performance of treating brines with different concentrations and compositions could provide some practical significance to verify membrane superhydrophobicity and that it is of significance to test low surface tension liquids such as surfactants or oils for wider range applications. In our work, we focus on the membrane performance of treating composition-complex real seawater and hypersaline brines (Fig. 7, Figs. S24, S25, S26, S27, Table S4). The results indicate the promising practical potential of our membranes. More importantly, we would like to clarify that our work primarily focuses on the following core novel aspects, including the proposed Janus membrane concept, molecular mechanism of enhanced water evaporation, superior performance and heat/mass transfer mechanism, and harsh operation stability, all of which have not been explored in previous works. In order to not defocus on these novel aspects, we would like to perform more detailed operational performance and stability tests of brines with low surface tension liquids in our followup work. Also, we tried to measure the sliding angle of brine on the membrane surface, but always failed due to the too high curvature of our micro-tubular membranes. We hope you could understand our situation and decision.

4. Are the test locations on thermal infrared images randomly selected? The authors should provide some details.

Response: Thank you for valuable comments and suggestions. To assure the accuracy, the test locations on thermal infrared images were randomly selected, but almost in the middle positions of radial direction of the membranes. To address your concerns, we have further provided some details (please see Page 22, revised manuscript), which are show as follows.

To probe membrane surface temperatures, thermal infrared pictures were randomly captured, which were almost in the middle positions of radial direction of the membranes, by an IR camera with a temperature precision of $\pm 2^{\circ}\text{C}$, measured range of $-20 - 280^{\circ}\text{C}$, IR resolution of 160×120 pixel, super IR resolution of 320×240 pixel and noise equivalent temperature difference of $< 0.12^{\circ}\text{C}$ (Testo 865, Testo Instrument International Trading Co. Ltd., China). The temperature was calibrated by a thermocouple at factory.

5. What is the filtration mode in this study? Dead-end or cross-flow?

Response: The filtration mode was cross-flow in this work. To address your concerns, we have further specified the cross-flow filtration mode by making minor revisions “in a lab-made cross-flow filtration mode membrane setup”. (please see Page 22, revised manuscript).

6. What is the difference between two membrane configurations the authors used, such as tube and flat-sheet?

Response: The difference between tubular and flat-sheet membrane configurations. To address your concerns, we have further specified the difference between tubular and flat-sheet membrane configurations by making some revisions. (please see Page 20, revised manuscript).

While the membranes in this work were designed in a tubular configuration with a higher packing density (i.e., higher membrane area per unit volume) than flat-sheet membranes, they may also be fabricated in flat-sheet configurations for more effective solar-driven interfacial desalination, where surface nanocarbon could more efficiently adsorb simulated sunlight (Figure S28, S29, Table S6).

7. The authors should clearly state the derivation of Equation 1 and cite relevant literature.

Response: Thank you for your suggestion. We must apologize that after careful discussion, we have decided to delete Equation 1 since we think that we have provided sufficient discussion on the effect of temperature and solar power density on water flux (Page 10, revised manuscript), which is also shown as follows.

Elevated feed temperature was also shown to improve desalination performance by enhancing the temperature gradient across the membrane and thereby accelerating water flux due to the improved water vapor pressure difference and thus driving force based on the Clausius Clapeyron equations and Fick's Law (Figure 4e, Figure S16). Similarly, enhancing solar power density could also improve water flux due to higher interfacial temperature while maintaining stable and high salt rejection.

8. Is the actual seawater pre-treated (microfiltration, etc.) to remove impurities such as particulate matter?

Response: The actual seawater was used without any pretreatment. To address your concerns, we have further specified this issue by making some revisions. (please see Page 22, revised manuscript).

Real seawater was obtained from Xinghai Park, China (38.87° N, 121.58° E) and used without any pretreatment.

9. The authors sampled real seawater as feed solution, but for solar-thermal membrane distillation, it would be more applicable if the developed membranes could be tested under actual sunlight.

Response: Thank you for this valuable suggestion. We do agree with your opinion that it would be more applicable if the developed membranes could be tested under actual sunlight. The focus of the current work is to investigate heat and mass transfer and interfacial evaporation mechanisms, as well as the desalination performance and stability of simulated saline waters with various compositions or real seawater for solar-thermal membrane distillation process. As you can see, we have provided solid data in characterization, performance and stability under various conditions, and detailed mechanism analysis via experimental and simulation protocols in this work. In order to not defocus on the current work, we intend to systematically investigate the performance and fundamental mechanistic insights under various conditions and under actual sunlight as a followup to the current work. We hope you can kindly understand our concerns and decision.

To address your concerns, we have further made some revisions (please see Page 20, revised manuscript), which are show as follows.

It would be more applicable that the developed membranes could be tested under actual sunlight, which is being investigated as a followup to the current work.

10. It seems that the permeate flux was slightly enhanced after turning on the light. The authors should emphasize the necessity of using this photothermal conversion.

Response: Thank you for this comment. Actually, the enhancement of permeate flux under different conditions is impressive before and after turning on light. For example, there are remarkable increase percents, for example, 13.35% and 32.17% (for 10 g L⁻¹ saline water), and 43.94% and 96.23% (for 35 g L⁻¹ saline water) at 1 and 3 KW m⁻² irradiation (Figure 4b). There are remarkable increase percents, ~24.8% (feed flow rate 2.2 L h⁻¹, Figure 4f) and ~36.28% (Figure 4g) before and after 1 KW m⁻² irradiation. For constant operation, there are remarkable increase percents 44.67% at the beginning sampling time and 261.95% at ending sampling time (seawater, Figure 7d), and 14.28% at the beginning sampling time and 378.70% at ending sampling time (gypsum-containing hypersaline brine, Figure 7e). Particularly, the enhancement of permeate flux is more significant for long-term operation (Figures 7d, 7e).

11. The authors provided detailed simulations on mass transfer and heat distribution, and also interesting mechanism analysis on interfacial water evaporation. This provides some useful fundamental scientific insights. The membrane showed good anti-scaling performance. In future, it is recommended to further elaborate the

anti-scaling mechanism of the as-developed membranes, especially the effects of their structural features and surface properties (superhydrophobicity, etc.) on the scaling behavior of the membrane surfaces.

Response: Thank you for this valuable suggestion. We do agree with your opinion that it would be necessary to further elaborate the anti-scaling mechanism of the as-developed membranes, especially the effects of their structural features and surface properties (superhydrophobicity, etc.) on the scaling behavior of the membrane surfaces. The focus of the current work is to investigate heat and mass transfer and interfacial evaporation mechanisms, as well as the desalination performance and stability of simulated saline waters with various compositions or real seawater for solar-thermal membrane distillation process. As you can see, we have provided solid data in characterization, performance and stability under various conditions, and detailed mechanism analysis via experimental and simulation protocols in this work. In order to not defocus on the current work, we intend to systematically investigate the performance and anti-scaling mechanism under various conditions as a followup to the current work.

To address your concerns, we have further made some revisions (please see Page 20, revised manuscript), which are show as follows.

It would be also interesting to systematically investigate the performance and anti-scaling mechanism under various conditions.

12. The section Discussion, should be a brief summary of the most important findings of the study rather than a repetitive description of the contents of the manuscript.

Response: We agree with your opinion that the section Discussion should be a brief summary of the most important findings of the study. To address your concerns, we have further made some revisions by summarizing the most important findings (please see Page 19-20, revised manuscript).

13. The format of references needs to be modified according to the requirement of the journal.

Response: Thank you for this good suggestion. We have carefully checked the format of all the references and revised to follow the rule of the journal Nature Communications.